



# Tomographic retrievals of ozone with the OMPS Limb Profiler: algorithm description and preliminary results

Daniel J. Zawada[1], Landon A. Rieger[1], Adam E. Bourassa[1], and Douglas A. Degenstein[1]

[1]University of Saskatchewan

*Correspondence to:* Daniel J. Zawada (daniel.zawada@usask.ca)

**Abstract.** Measurements of limb scattered sunlight from the Ozone Mapping and Profiler Suite Limb Profiler (OMPS-LP) can be used to obtain vertical profiles of ozone in the stratosphere. In this paper we describe a two-dimensional, or tomographic, retrieval algorithm for OMPS-LP where variations are retrieved simultaneously in altitude and the along orbital track dimension. The algorithm has been applied to measurements from the center slit for the full OMPS-LP mission to create the publicly available USask OMPS-LP 2D v1.0.2 dataset. Tropical ozone anomalies are compared with measurements from the Microwave Limb Sounder (MLS) where differences are less than 5% of the mean ozone value for the majority of the stratosphere. Examples of near coincident measurements with MLS are also shown, and agreement at the 5% level is observed for the majority of the stratosphere. Both simulated retrievals and coincident comparisons with MLS are shown at the edge of the polar vortex, comparing the results to a traditional one-dimensional retrieval. The one-dimensional retrieval is shown to consistently overestimate the amount of ozone in areas of large horizontal gradients relative to both MLS and the two-dimensional retrieval.

## 1   Introduction

The Ozone Mapping and Profiler Suite Limb Profiler (OMPS-LP) on-board the Suomi-NPP spacecraft began taking routine measurements of limb scattered sunlight in early April 2012 (Flynn et al., 2006). The limb profiler images the atmospheric limb every 19 s ($\sim$125 km along track) from the ground to approximately 100 km using three vertical slits that are separated horizontally by $4.25°$. A prism disperser is used to obtain a spectrally resolved signal in the range 290–1000 nm. These spectrally resolved measurements can be inverted with a forward model accounting for multiple scattering to obtain vertically resolved profiles of ozone concentration in the atmosphere.

The standard OMPS-LP ozone product is produced by NASA and version 1.0 of the retrieval is described in detail by Rault and Loughman (2013). The NASA retrieval employs the assumption of horizontal homogeneity, treating each vertical image separately to retrieve a one-dimensional vertical profile. However, it is possible to take advantage of the long limb path length and fast sampling capabilities of OMPS-LP to combine multiple images together and retrieve in the orbit track and altitude dimensions simultaneously. These two-dimensional, or tomographic, retrievals have been used successfully in many retrievals from limb emission instruments (e.g. Degenstein et al., 2003, 2004; Livesey et al., 2006; Carlotti et al., 2006). A two-dimensional retrieval of $NO_2$ was done for limb scatter measurements from the SCanning Imaging Absorption SpectroMeter for Atmospheric CHartographY (Bovensmann, 1999) by Puķīte et al. (2008) and a preliminary two-dimensional retrieval study





for ozone using a single scatter radiative transfer model was also performed by Rault and Spurr (2010) using simulated OMPS-LP measurements. Measurements from OMPS-LP are a natural candidate to attempt a two-dimensional retrieval due to the relatively fast along orbital track sampling ($\sim$125 km) compared to other limb scatter instruments, for example, the Optical Spectrograph and InfraRed Imaging System (OSIRIS) (Llewellyn et al., 2004) has $\sim$600 km along track sampling.

In this paper we describe a retrieval algorithm for the central slit of OMPS-LP which accounts for inhomogeneity in the along orbit direction, and present preliminary results. To the author's knowledge this is the first two-dimensional limb scatter ozone retrieval applied to real measurements. The algorithm is described in detail in Sec. 2. We have applied the algorithm to the entire mission of OMPS-LP, creating a dataset of vertical profiles of stratospheric ozone from early 2012 to present with near global coverage (USask OMPS-LP 2D v1.0.2 dataset). Section 5 presents some preliminary results and validation efforts

with the dataset. Tropical ozone anomalies are compared against those from the Microwave Limb Sounder (MLS) (Waters et al., 2006) for the full mission dataset. Lastly, nearly perfect coincident measurements with MLS are investigated.

## 2   The retrieval algorithm

### 2.1   Overview

Here we follow the optimal estimation framework outlined in Rodgers (2000) and use similar notation. The general goal of the

atmospheric inverse problem is to find the optimal set of state parameters, $\boldsymbol{x}$, given with a set of measurements, $\boldsymbol{y}$, and other apriori information or constraints. The vector $\boldsymbol{x}$ of length $n$ is often called the state vector, while the vector $\boldsymbol{y}$ of length $m$ is called the measurement vector. One approach to this problem is to minimize the cost function,

$$\chi^2 = [F(\boldsymbol{x}) - \boldsymbol{y}]^T S_\epsilon^{-1} [F(\boldsymbol{x}) - \boldsymbol{y}] + [\boldsymbol{x}_a - \boldsymbol{x}]^T R^T R[\boldsymbol{x}_a - \boldsymbol{x}], \tag{1}$$

where $\mathbf{S}_\epsilon$ is the covariance of the measurement vector, $\mathbf{F}$ is the forward model, $\mathbf{R}$ is a regularization matrix, and $\boldsymbol{x}_a$ is the

apriori state vector. Apriori information is included through the two quantities $R$ and $x_a$. Applying a standard Guass-Newton minimization approach to the cost function results in the iterative step,

$$\boldsymbol{x}_{i+1} = \boldsymbol{x}_i + (\mathbf{K}_i^T \mathbf{S}_\epsilon^{-1} \mathbf{K}_i + \mathbf{R}^T \mathbf{R} + \gamma_i \mathbf{I})^{-1} \cdot$$
$$\left[ \mathbf{K}_i^T \mathbf{S}_\epsilon^{-1} (\boldsymbol{y} - \mathbf{F}(\boldsymbol{x}_i)) - \mathbf{R}^T \mathbf{R}(\boldsymbol{x}_i - \boldsymbol{x}_a) \right], \tag{2}$$

where $\mathbf{K}$ is the Jacobian matrix of the forward model, $i$ is the iteration number, and $\gamma_i$ is a Levenberg-Marquardt damping parameter. The Levenberg-Marquardt type regularization term $\gamma_i I$ may be included to move the solution step closer to that of

a gradient descent method, aiding performance when the Gauss-Newton step is outside the linear regime. Equation (2) forms the basis of the retrieval method used in this work.

The tomographic or two-dimensional nature of the retrieval is encoded in the details of the definitions of the state vector and the measurement vector. The state vector contains information about the atmospheric state for an entire orbit of OMPS-LP and is described in detail in Sec. 2.2. A brief description of the forward model, which must account for atmospheric variations




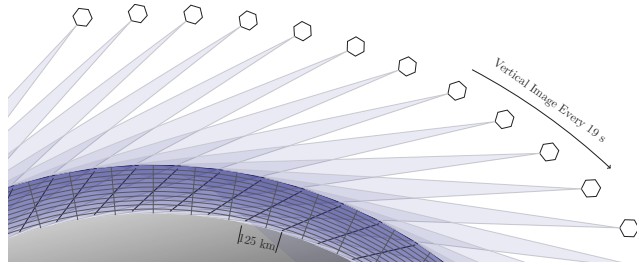

**Figure 1.** Conceptual image (not to scale) of the OMPS-LP viewing geometry and retrieval grid. The retrieval grid locations (white gray lines) are chosen to match the average tangent point of the OMPS-LP measurements (solid black lines).

along the line of sight, is given in Sec. 2.3 and Sec. 2.5. The form of regularization and apriori used is given in Sec. 2.6. Lastly, the form of the measurement vector for each retrieved species is presented in Sec. 2.7.

For this work v2.0–2.4 of the OMPS-LP L1G product (https://ozoneaq.gsfc.nasa.gov/data/omps/) is used.

## 2.2 The state vector

The state vector is a representation of the atmospheric state on a discrete grid, which we will refer to as the retrieval grid. The retrieval grid consists of a set vertical profiles at discrete locations along the orbital track of the instrument's tangent point. Between grid points bi-linear interpolation is applied to create a continuous representation of the atmosphere. The state vector for a single trace species is a flattened representation of this grid, with altitude being the leading dimension. As OMPS-LP measures scattered sunlight, each orbit has a natural start and stop point characterized by high solar zenith angles. In

constructing the retrieval grid we use images with solar zenith angles at the tangent point of less than $88°$.

Figure 1 shows an illustration of the constructed retrieval grid and its relation to measurements from OMPS-LP. The horizontal resolution of the retrieval grid is chosen to match the horizontal sampling of OMPS-LP, which is approximately $125\,\mathrm{km}$. A consequence of the limb viewing geometry is that measurements with a higher tangent point are closer to the instrument than measurements with a lower tangent point. For convenience the absolute locations of the horizontal retrieval grid locations

(white gray lines in Fig. 1) is chosen to match the average tangent point of each measurement image.

A consequence of performing a tomographic retrieval is that there is less information at the edges of the retrieval grid, simply because there are less measurements which sample near the edges. A common approach to minimize this effect is to cut off the ends of the retrieval, e.g. retrieve on a grid from $0°$ to $100°$ but only report the results for $5°$ to $95°$. This approach works well when the next retrieval starts where the first one ended, here one may allow some overlap between the two retrievals, throwing

out both edges and merging the results together. The final result would be nearly eliminating the edge effects for a small cost in increased computational time. However, if the edge is a physical limit for the retrieval, as is the case for this retrieval, then cutting off the ends of the retrieval will result in a loss of data.

In order to maximize the amount of data retrieved in the OMPS-LP retrieval, we use a similar but different approach. As previously mentioned, our retrieval grid has hard cutoffs at solar zenith angle $88°$. However, when constructing the measurement



vector we use all images with solar zenith angle less than $90°$. Under this approach we have not noticed unphysical effects at the edges of the retrieval. However there is still less information present at the retrieval boundaries, which is reflected in the resolution and precision estimates described in Sec. 4. The latitudinal coverage of OMPS-LP, and thus the retrieval grid, varies throughout the course of the year as the illuminated portion of the Earth changes. The latitude region $60°$ S to $60°$ N is sampled

near continuously throughout the year, while coverage extends to $82°$ in each hemisphere's summer.

## 2.3   The forward model

The forward model used in this study is SASKTRAN-HR (Bourassa et al., 2008; Zawada et al., 2015). SASKTRAN-HR solves the radiative transfer equation in integral form using the method of successive orders initialized with the incoming solar irradiance. The model is capable of handling inhomogeneities in the atmospheric state in the along line of sight direction. In

addition to radiance, the model also outputs the Jacobian matrix with respect to the underlying two-dimensional atmosphere. Jacobians are calculated analytically taking into account all first order of scatter terms with approximations made for higher order terms. The forward model and the Jacobian calculation are described in depth in Zawada et al. (2015) and Zawada et al. (2017) respectively.

   Due to the Earth's rotation, there is a slight mismatch between the line of sight plane and the retrieval grid as is shown in

Fig.2. The horizontal distance between the next image's average tangent point and the previous image's line of sight plane is approximately $\sim$10 km near the equator, with the effect diminishing near the poles. Since SASKTRAN-HR specifies the atmosphere in the line of sight plane, some transformations need to be performed during the retrieval process. At the beginning of each iteration, the atmosphere specified on the retrieval grid is transformed to the internal SASKTRAN-HR representation. The radiative transfer calculation is then performed, obtaining both the radiance and the Jacobian matrix. Since the Jacobian

matrix was calculated on the internal SASKTRAN-HR atmosphere grid, this needs to be transformed back to the retrieval grid representation. These transformations are typically quite small in effect, and are done taking into account the symmetries that SASKTRAN-HR assumes in the radiative transfer calculation.

## 2.4   Computational considerations

In a tomographic retrieval, the length of the state vector, $n$, and the length of the measurement vector, $m$, are significantly

larger than those of a one-dimensional retrieval. For example, if the retrieval grid was set up to match the inherent resolution of the OMPS-LP measurements of a single orbit, for each species $n$ would be on the order of 10000, and for each wavelength $m$ would also be on the order of 10000. Storing these vectors does not pose any computational challenge, however, it quickly becomes necessary to store the $m \times n$ Jacobian matrix using sparse storage techniques. The Jacobian matrix is naturally sparse in the horizontal direction as sensitivity is largest at the tangent point and decreasing away from it. For the limb scattering

problem involving multiple scattering elements of the Jacobian matrix are never truly zero, every point in the atmosphere should in theory contribute to every measurement. However, owing to the approximations made in the Jacobian calculation outlined in the section prior, contributions are only calculated along the line of sight and solar planes resulting in a sparsity



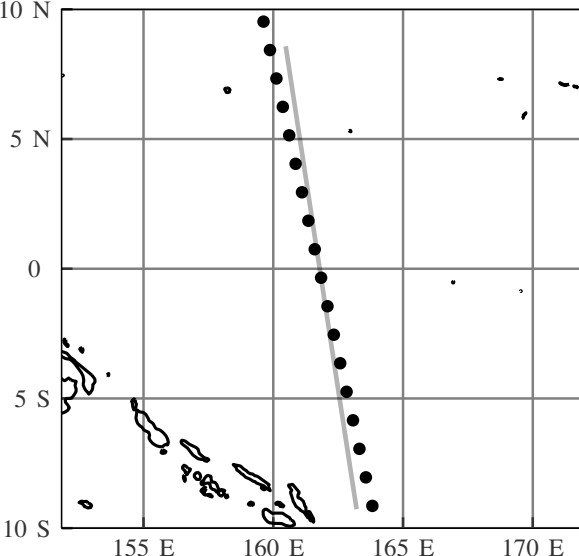

**Figure 2.** Example of mismatch between the line of sight plane and the tangent point ground track. Black dots show the tangent points (at 25 km) for OMPS-LP orbit 14940. The gray line represents the line of sight plane for the tangent point intersecting the line.

factor of $\sim0.05$. The sparsity of the Jacobian matrix can be improved by artificially allowing only profiles less than some specified distance to the tangent point to contribute, as is done in Livesey et al. (2006).

While every matrix in Eq.( 2) is sparse, it is often desirable from a computational speed point of view to store some combinations of matrices densely. In particular, solving the linear system requires computing the $n \times n$ $(\mathbf{K}_i^T \mathbf{S}_\epsilon^{-1} \mathbf{K}_i + \mathbf{R}^T \mathbf{R} + \gamma_i \mathbf{I})$

matrix. While still somewhat sparse, we have observed significant speed increases by solving the linear system densely. For a full OMPS-LP orbit this matrix would be $10000 \times 10000$, taking less than $1\,\mathrm{GB}$ of memory.

## 2.5   Accounting for the time dependence

Due to an inadequate amount of measurements we do not account for the time variation of the ozone field in the retrieval. The reported time for each retrieved profile is calculated by interpolating the measurement times on the tangent points to the

retrieval grid. While not perfect, we expect this is a good estimate as the majority of information for a single retrieved profile originates from the images that have tangent points near it. Nevertheless, there are several other time dependent effects which play a role in how the retrieval is performed.

The radiative transfer equation is explicitly time dependent owing to the changing solar conditions. For an imaging instrument such as OMPS-LP, the natural and most accurate solution to this problem is to re-run the forward model for every image.

That being said, there is potential for large computational speed improvements by combining multiple images into the same forward model calculation. Since SASKTRAN-HR solves the radiative transfer equation in a region of interest (nominally a



$10°$ cone, but this can be configured) around the tangent point, there is considerable overlap between the field of interest of one image to the next. However, there are issues in performing this combination:

1. Each image happens at a different instant in time, thus the solar conditions have changed.

2. SASKTRAN-HR's internal atmosphere is specified as a plane. The lines of sight from one image do not necessarily lie in the same plane as the lines of sight for the next image. Furthermore, the more images that are combined together the larger this plane will diverge from the retrieval grid.

3. The Earth is represented internally as a sphere with curvature matching a reference ellipsoid at the average tangent point which changes from image to image.

The first and the third conditions are not unique to tomographic retrievals, limb scanning instruments face similar challenges in one-dimensional retrievals. For example, a single OSIRIS limb scan sequence takes approximately $90\,\mathrm{s}$ and is modeled with a single forward model calculation in operational retrieval algorithms (Degenstein et al., 2009). Internal tests have been performed to quantify the three conditions by comparing results that modeled every image separately with retrievals that combined 5 subsequent images together ($95\,\mathrm{s}$ variation from the first image to the last image) which resulted in mostly random differences in retrieved ozone on the order of $0.5\%$.

## 2.6 Regularization

Most atmospheric retrieval methods fall into two classes, the first is where no regularization term ($R$ in Eq. (2)) is used. In this case the resolution of the retrieved profile is determined by how the state vector is defined, i.e., the resolution of the retrieval grid. It is always desired to make the retrieval grid as fine as possible, but one has to be careful as to not make the grid too fine, allowing over-fitting. In the one-dimensional case retrievals that operate without regularization commonly choose the vertical grid to match the sampling resolution of the instrument. A large advantage of this approach is that the retrieval resolution stays constant over time, side-stepping problems when looking for small changes in a long time series.

The second approach is to make the retrieval grid finer than the expected resolution of the retrieval, and include some form of regularization in Eq. (2). Regularization acts to include some additional information in the retrieval. If the added information is correct, then the retrieval results in the optimal solution, however this is rarely the case. The additional information could be in the form of an a priori atmospheric state with associated covariance, or it could take the form of an ad-hoc constraint. Common constraints are to impose a cost on the first or second derivative of the state vector. While these constraints do not have a formal justification, they do have some physical basis in that it is expected the retrieved profile be continuous.

For our retrieval we use a combination of both of the above methods. Since the vertical direction is typically what limb geometry measurements are designed to target, we apply a second derivative constraint only in the horizontal direction of the





retrieval grid. The regularization matrix takes the form,

$$
R = \alpha \begin{pmatrix}
-\frac{1}{4} & \mathbf{0} & \frac{1}{2} & \mathbf{0} & -\frac{1}{4} & 0 & 0 & \dots \\
0 & -\frac{1}{4} & \mathbf{0} & \frac{1}{2} & \mathbf{0} & -\frac{1}{4} & 0 & \dots \\
& 0 & -\frac{1}{4} & \mathbf{0} & \frac{1}{2} & \mathbf{0} & -\frac{1}{4} & \dots \\
\vdots & \vdots & \vdots & \vdots & \vdots & \vdots & \vdots & \ddots
\end{pmatrix},
\tag{3}
$$

where $\alpha$ is a constant scaling factor used to control the amount of regularization and $\mathbf{0}$ indicates a number of zeros equal to the number of altitude grid points. For simplicity the apriori state vector of Eq. (2) is chosen to be 0. As the regularization matrix used only applies in the horizontal direction, the horizontally integrated vertical resolution of the retrieved profiles matches the vertical resolution of the retrieval grid.

It should be noted that even though we apply no constraints in the vertical direction, the retrieval software is capable of doing so. While the above discussion treats the horizontal and vertical dimensions of the grid as separate entities, their resolutions are inherently coupled together. A lower resolution horizontal grid allows for a higher resolution vertical grid, keeping the total information content relatively constant, and vice-versa. We make no claims on what is the optimal relationship between these two resolutions, and it is something that we are actively investigating. It is important to mention that a one-dimensional retrieval makes the trade-off decision for you, allowing control of only the vertical constraint. The effects of a one-dimensional retrieval on horizontal regularization has been studied for the Michelson Interferometer for Passive Atmospheric Sounding by von Clarmann et al. (2008).

## 2.7 Retrieval ordering

The retrieval is performed for three major parameters: ozone number density, stratospheric aerosol number density, and surface reflectance assuming a Lambertian surface. While considerable effort has been put into both the aerosol and surface reflectance retrievals, they are performed primarily as a second order correction for the ozone retrieval. Each species is retrieved independently, i.e., holding the other parameters fixed; but the overall retrieval operates in stages, feeding the results of previous parameter retrievals into the current one. The general retrieval order follows that of Degenstein et al. (2009) and is first surface reflectance, then aerosol number density, and then lastly ozone number density. Two passes of this overall procedure are performed, allowing results from the ozone retrieval to couple back into the other retrievals. The first pass of the procedure can be thought of as obtaining a good first guess for state vectors, while the second pass finalizes the retrieval.

A fixed number of iterations is performed in each of the passes. The first round of the retrieval procedure performs five iterations for each of the targeted quantities while the second round performs two iterations. Various diagnostic information is also calculated, including the normalized $\chi^2$ value and the expected $\chi^2$ value at the next step assuming the problem is linear. At the end of the fixed number of iterations it was found that these two values always match within $1\%$, indicating that the solution has likely converged. It is planned for a future version of the retrieval software to stop early if convergence is detected, however this is not expected to improve the solution only the computational efficiency.





| Ozone Sensitive Wavelength [nm] | Reference Wavelength(s) [nm] | Valid Altitudes [km] |
|---|---|---|
| 292.43 | 350.31 | 22–59 |
| 302.17 | 350.31 | 22–55 |
| 306.06 | 350.31 | 22–51 |
| 310.70 | 350.31 | 22–48 |
| 315.82 | 350.31 | 22–46 |
| 322.0 | 350.31 | 22–42 |
| 331.09 | 350.31 | 22–39 |
| 602.39 | 543.84, 678.85 | 0–30 |

**Table 1.** Wavelength triplet/doublets used in the ozone retrieval.

### 2.7.1 Ozone

The ozone retrieval uses a common technique first suggested by Flittner et al. (2000) where ozone sensitive wavelengths in the Hartley-Huggins and Chappuis bands are normalized by both ozone insensitive wavelengths and high altitude measurements. This technique, sometimes referred to as the triplet or doublet method, has been used successfully in a variety of limb scatter ozone retrievals (e.g. von Savigny et al., 2003; Loughman et al., 2005; Rault, 2005; Degenstein et al., 2009; Rault and Loughman, 2013). The ozone cross section used in the retrieval is compiled from Daumont et al. (1992); Brion et al. (1993); Malicet et al. (1995). While the triplet/doublet method has previously only been implemented for one-dimensional retrievals, many of the ideas are still applicable to two-dimensional retrievals with some modifications.

Our ozone measurement vector consist of 7 doublets in the Hartley-Huggins absorption bands and one triplet in the Chappuis absorption band shown in Table 1. In one-dimensional retrievals the UV doublets are often forced to only contribute when the atmosphere is optically thin, i.e. when the area of maximal sensitivity is at the tangent point. This can be done through either analyzing the diagonal elements of the Jacobian matrix directly (Loughman et al., 2005), or by only using altitudes above the "knee" of the atmosphere as is done in Degenstein et al. (2009). The primary reason to do this forcing is so that the retrieval is most sensitive to the tangent point, to minimize the effect of the implicit horizontal homogeneity assumption. Since the assumption of horizontal homogeneity is broken for the tomographic retrieval, we allow all UV doublets to contribute down to some minimum altitude, chosen to be $22\,\mathrm{km}$. This altitude is approximately the knee of the 350 nm radiance profile, as seen in Fig. 3, radiances below this altitude are heavily sensitive to atmospheric upwelling and in particular absorbing aerosols.

The unnormalized measurement vector, $\tilde{y}$, is given by,

$$\tilde{y}_{jkl} = \frac{1}{n_{\mathrm{ref}_l}} \sum_{\lambda \in \mathrm{ref}_l} \log\left[I_j(h_k, \lambda)\right] - \frac{1}{n_{\mathrm{sens}_l}} \sum_{\lambda \in \mathrm{sens}_l} \log\left[I_j(h_k, \lambda)\right], \tag{4}$$

where $j$ indexes image along an orbit, $k$ indexes tangent altitude, $l$ indexes the triplet, and the sets $\mathrm{ref}_l$ and $\mathrm{sens}_l$ are the reference and sensitive wavelengths for triplet $k$ from Table. 1 with corresponding lengths $n_{\mathrm{ref}_k}$ and $n_{\mathrm{sens}_k}$ respectively. Each triplet/doublet is normalized by its value at high altitude where the ozone sensitivity is minimal. The high altitude normali-





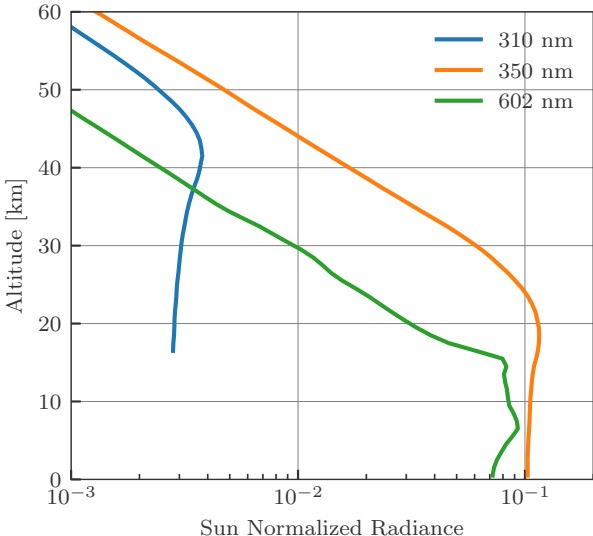

**Figure 3.** Sun normalized radiances observed by OMPS-LP event number 90 of orbit 19490.

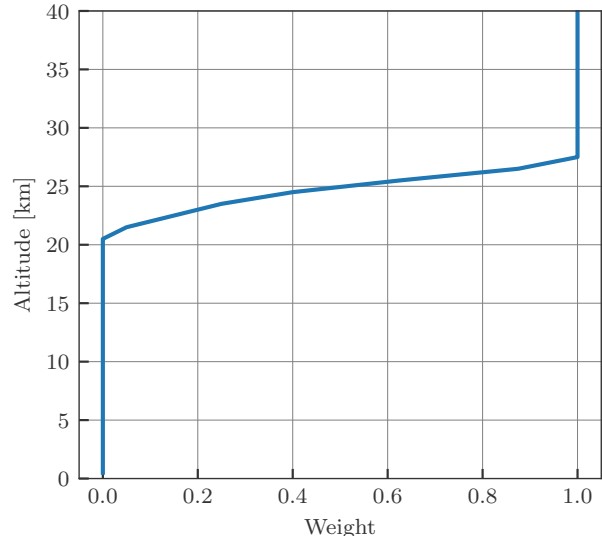

**Figure 4.** Scaling factors as a function of altitude applied to the UV doublet measurement error covariances.

zation helps to minimize any errors in the absolute calibration of the instrument. The normalization altitude varies for each doublet/triplet (shown in Table 1), and is pushed low to minimize stray light errors.





To avoid discontinuities caused by suddenly introducing UV triplets near $22\,\mathrm{km}$, the diagonal of the measurement error covariance matrix is artificially scaled during the retrieval,

$$S_{\epsilon,ii} = \frac{S_{\epsilon,ii}}{w^2}, \tag{5}$$

where the weights, $w$, are only applied to the UV triplets and only depend on altitude. The applied scale factors are shown in
Fig. 4.

### 2.7.2 Stratospheric aerosol

The stratospheric aerosol measurement vector definition follows closely the one outlined in Bourassa et al. (2012) and applied to OSIRIS measurements, with a few minor modifications. The unnormalized measurement vector is given by,

$$\tilde{y}_{jk} = \log\left[I_j(h_k, 745.67\,\mathrm{nm})\right] - \log\left[I_{j,\mathrm{ray}}(h_k, 745.67\,\mathrm{nm})\right], \tag{6}$$

where $I_{\mathrm{ray}}$ is a radiative transfer calculation performed with no aerosols in the atmosphere (pure Rayleigh background). Similar to the ozone retrieval the measurement vector is normalized by a high altitude measurement. The altitude of normalization is chosen following the technique described by Bourassa et al. (2012) where the normalization altitude is adjusted on an image by image basis to minimize effects of stray light. Adjusting the normalization altitude on an image by image basis can cause sharp jumps in the normalization altitude across the orbital direction. It is currently unclear whether or not this adjustment is
ideal for a two-dimensional retrieval, however as the aerosol retrieval is done primarily as a first order correction for the ozone retrieval this has not been investigated in detail.

The measurement vector described here differs from that of Bourassa et al. (2012) in that there is no normalization relative to a shorter wavelength (470 nm for the OSIRIS retrieval). The short wavelength normalization was included to reduce the dependence of knowledge of the background Rayleigh atmosphere. However issues were encountered in that the short wave-
length would often be measured on a different gain setting than the longer wavelength, introducing artifacts in the retrieval (see Jaross et al., 2014, for more information on the gain settings of OMPS-LP). Since there exist many limb scatter aerosol retrieval algorithms that operate without a short wavelength normalization Rault and Spurr (2010, e.g.) for simplicity we have opted to remove it. Stratospheric aerosols in the retrieval are assumed to consist of liquid $H_2SO_4$ spherical droplets following a log-normal particle size distribution with a median radius of $80\,\mathrm{nm}$ and a mode width of 1.6. The phase function is calculated
using a standard Mie scattering code (Wiscombe, 1980).

### 2.7.3 Albedo

The forward model assumes a Lambertian reflecting surface parameterized by the albedo, the ratio of outgoing to incoming radiance. Typically this quantity does not physically represent actual reflectance from the surface of the Earth, but is used as an approximation for all upwelling from the troposphere. It is important to retrieve the albedo as many wavelengths used in the
ozone retrieval are affected by atmospheric upwelling.





While the albedo is handled in a two-dimensional sense in the forward model, several assumptions are made which make the albedo retrieval similar to a set of independent one dimensional retrievals. Furthermore the albedo retrieval is not done under the Rodgers approach described earlier, and instead follows the approach of Bourassa et al. (2007). We define the albedo state vector $x_{\mathrm{alb}}$ as the albedo on the surface of the Earth assuming a Lambertian surface at a set of latitudes and longitudes defined by the 40 km tangent point of each image. Therefore the state vector is the same length as the number of images used in the retrieval.

The albedo is iteratively updated with the equation,

$$x_{\mathrm{alb},j}^{i+1} = x_{\mathrm{alb},j}^{i} \frac{I_{j,\mathrm{meas}}(40\,\mathrm{km}, 745.67\,\mathrm{nm})}{I_{j,\mathrm{mod}}(40\,\mathrm{km}, 745.67\,\mathrm{nm})}. \tag{7}$$

The measurement vector uses the same wavelength as the aerosol retrieval since their effects tend to be coupled together. The retrieval is one-dimensional in the sense that, at least for one specific iteration, each image is allowed to only affect one element of the albedo state vector. However the forward modeled radiance is calculated using the two-dimensional albedo field, which allows images to couple to other elements of the state vector over the course of multiple iterations.

## 3 Pointing correction

Accurate and stable pointing knowledge is of particular importance for limb-scatter measurements as it is typically not possible to simultaneously measure pressure. Moy et al. (2017) provides a detailed characterization of the OMPS-LP pointing errors, however many of these corrections have not been applied to the v2.0–2.4 L1G product used in this study. Therefore for the current version of the retrieval a separate pointing analysis has been performed.

We apply the Rayleigh Scattering Attitude Sensor (RSAS) (Janz et al., 1996) to the OMPS-LP measurements. The ratio of the measured radiance at 40 km and 20 km near 350 nm is compared to the calculated radiance. At 40 km the radiance is sensitive to tangent altitude changes, while at 20 km the radiance is not very sensitive since the line of sight path has become optically thick. Based on the difference between the measured and modeled ratios it is possible to calculate an effective tangent altitude offset. The RSAS technique is sensitive to both atmospheric upwelling and stratospheric aerosol loading which makes it difficult to apply at low solar zenith angles and in forward scatter conditions respectively.

To minimize the effects of both atmospheric upwelling and stratospheric aerosols we only use measurements with solar zenith angle between $70°$ and $50°$ with solar scattering angles greater than $90°$. Measurements satisfying similar criteria have recently been successfully used to apply an RSAS pointing correction to OSIRIS retrievals by Bourassa et al. (2017). While measurements with a solar zenith angle greater than $70°$ would have even less upwelling, it is more challenging to accurately model the multiple scatter component of the radiance. Cutoffs greater than $50°$ were not found to affect the results, $50°$ was chosen to maximize the number of measurements. The altitude offsets were daily averaged and shown in Fig. 5. Offsets range from approximately 400 m to 0 m with a clear seasonal cycle, in April 2013 there is noticeable $\sim100$ m drop due to a known star tracker adjustment. Being able to clearly observe the star tracker adjustment provides confidence that at least on a relative scale we are able to detect pointing shifts with the RSAS method.





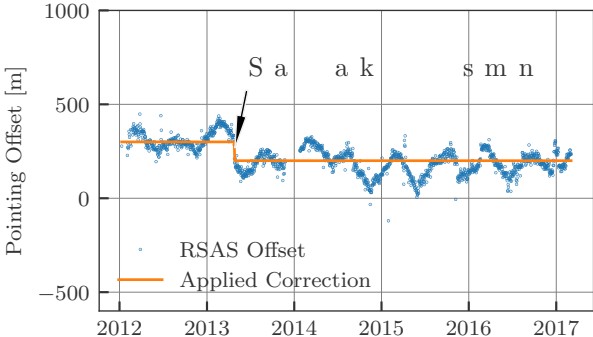

**Figure 5.** Daily averaged pointing offsets calculated with the RSAS technique. The orange line shows the applied pointing correction for v1.0.2 of the retrieved data product.

It is currently unknown whether or not the seasonal structure represents a true pointing shift or if it is an artifact of the RSAS method–perhaps due to the average latitude of the measurements also varying seasonally and changing cloud cover. We do not detect any significant pointing drift greater than $\pm 100\,\mathrm{m}$, however later years are affected by stratospheric aerosols from Kelud and Calbuco which may skew the RSAS technique. Preliminary validation efforts have revealed that, on average, there is likely

an absolute pointing error present in the OMPS-LP measurements. To calculate the applied pointing correction (solid line in Fig. 5) we take an average value both before and after the star tracker adjustment. All ozone profiles are shifted downwards by this amount after the retrieval has been performed. It should be noted that this applied pointing correction is by intention simple. A future version of the data product will examine the pointing in more depth, and apply the correction to the instrument lines of sight rather than post shifting the retrieved profile.

**4   Error analysis and resolution**

Both the random and systematic error components of a limb scatter ozone retrieval algorithm for a similar, but one-dimensional, retrieval have been studied in Loughman et al. (2005). Applying the conclusions of Loughman et al. (2005) to our retrieval algorithm suggests that the dominant sources of random error are pointing knowledge and the error due to measurement noise. Rault and Loughman (2013) have also presented similar findings for a one-dimensional retrieval algorithm applied to OMPS-

LP, and in particular showed that the error due to measurement noise is representative of the total random error budget. We will not repeat these analyses here, rather we will simply present the technique used to calculate the reported error estimate for each orbit.

Under the Rodgers framework the gain matrix, $\hat{\mathbf{G}}$, is given by,

$$\hat{\mathbf{G}} = (\hat{\mathbf{K}}^T \mathbf{S}_\epsilon^{-1} \hat{\mathbf{K}} + \mathbf{R}^T \mathbf{R})^{-1} \hat{\mathbf{K}}^T \mathbf{S}_\epsilon^{-1}, \tag{8}$$





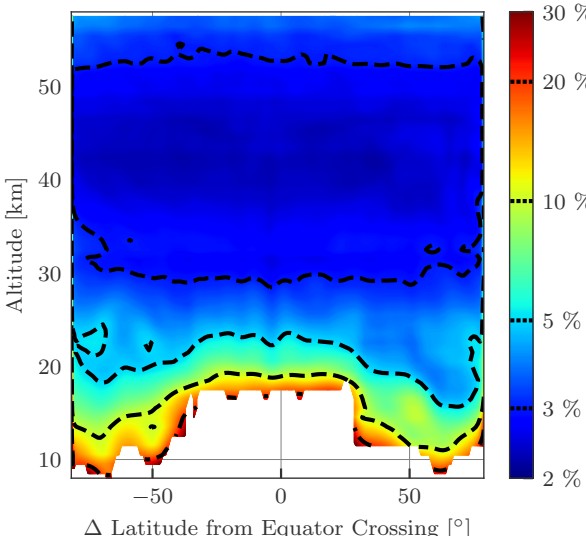

**Figure 6.** Precision estimate for ozone in percent for OMPS-LP orbit 27695 (March 2nd, 2017, 10:30 AM UTC at equator crossing). Contour levels are indicated by dashed lines on the colorbar. The corresponding retrieved ozone profiles are shown in Fig. 9.

and the averaging kernel,

$$\mathbf{A} = \hat{\mathbf{G}}\hat{\mathbf{K}},$$ (9)

where the hats indicate that the solution has converged. The solution covariance due to measurement noise only can also be estimated as,

$$\hat{\mathbf{S}}_{\mathrm{noise}} = \hat{\mathbf{G}}\mathbf{S}_\epsilon\hat{\mathbf{G}}^T.$$ (10)

In the current version of the retrieval only the solution covariance due to measurement noise is reported. For the purposes of the precision estimate we assume that the measurement covariance is diagonal, with the radiance measurements having a signal to noise ratio of 100, an upper bound on the error estimate taken from Jaross et al. (2014). Only the diagonal elements of the solution covariance are used for the error estimate. Since the state vector is the logarithm of number density, the precision
estimate in logarithmic space is propagated to linear space for the reported precision estimate.

Figure 6 shows an example precision estimate for OMPS-LP orbit 27695 (March 2nd, 2017, 10:30 AM UTC at equator crossing). Precision estimates are in the range 2–5% for the majority of the middle and upper stratosphere. In the lower stratosphere precision is ∼10% increasing to 30% near the tropopause. Various edge effects of the retrieval are also visible, most noticeably the increase in error at the beginning and end of the orbit but near where the tropopause lowers at mid-latitudes.
These are expected effects, edges of the retrieval grid inherently have less measurements contributing to them, increasing the expected noise. The estimate precision varies only slightly between orbits, and the values stated above are generally valid for the entire dataset.





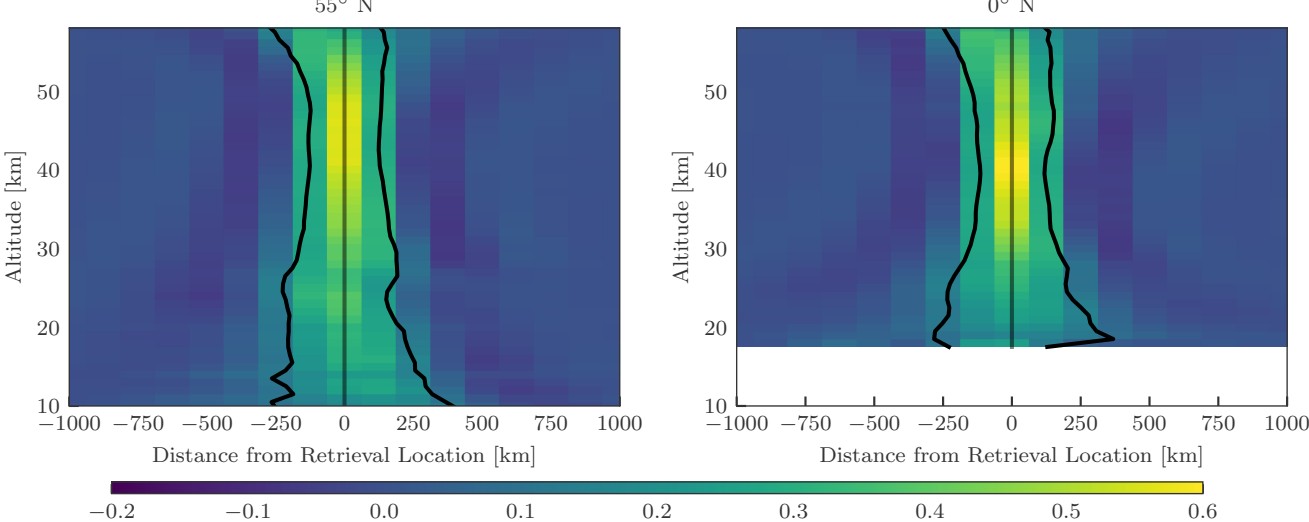

**Figure 7.** Horizontal averaging kernels from OMPS-LP orbit 27695 (March 2nd, 2017, 10:30 AM UTC at equator crossing) for $55°$ N (left panel) and $0°$ N (right panel). Data is masked below the lowest retrieval altitude. Vertical black lines show the full width at half maximum boundaries, while the vertical gray line indicates the location of the retrieval.

The resolution of the retrieval is found by analyzing the retrieval averaging kernels. As the retrieval is two-dimensional, each row of the averaging kernel contains both vertical and horizontal components. Since the regularization term (Eq. (3)) contains no vertical information, it can be shown that the horizontally summed averaging kernel (i.e., the vertical averaging kernel) is the identity matrix. This has also been verified by calculating the vertical averaging kernel for a set of OMPS-LP orbits, which

were all found to be identically unity. Considering the vertical averaging kernels contain little information, we will focus on the vertically integrated, or horizontal, averaging kernels.

Figure 7 shows the horizontal averaging kernel for orbit 27695 at $0°$ N and $55°$ N. In both cases the horizontal FWHM is smallest near 40 km with values of $\sim 250$ km. For the majority of the altitude range the FWHM is less than $400$ km, with the exception of the region near the tropopause where it can increase to $500$ km. Only minor differences are seen between the

tropical and mid-latitude averaging kernels, with the majority of the differences occuring near the lower bound of the retrieval. Averaging kernels are not stored for every orbit due to size constraints, however it was found that there was almost no variation from orbit to orbit and the ones shown here are representative for the entire dataset.

As previously stated, the vertical averaging kernels are identity, suggesting that the vertical resolution of the retrieval is 1 km, the same as the retrieval grid. However, the instrumental vertical field of view ($\sim 1.5$ km, see Jaross et al., 2014) is neglected

in the retrieval process, treating each measurement with a single line of sight. Therefore we estimate the vertical resolution of the retrieved profiles to be 1–2 km. It is intended to investigate the vertical resolution in more detail for future versions of the retrieval.



## 5    Preliminary results

### 5.1    Simulations on the edge of the polar vortex

To test the retrieval method, a one-dimensional retrieval method that assumes horizontal homogeneity has also been developed to compare against. The one-dimensional retrieval has been designed to be as similar to the two-dimensional retrieval as possi-

ble. The measurement vectors for ozone, albedo, and stratospheric aerosol are the same as those those for the two-dimensional retrieval, with the only difference being that the number of images used is one instead of an entire orbit. The state vector is modified to be one-dimensional in altitude, representing a horizontal homegenous atmosphere with $1\,\mathrm{km}$ vertical spacing. As the Tikhonov regularization is only applied in the horizontal direction for the two-dimensional retrieval, no regularization is used in the one-dimensional retrieval. The same iterative procedure is also used for the one-dimensional retrieval.

To test the ability of the two-dimensional retrieval to resolve horizontal gradients in the ozone field, simulated retrievals have been performed. For the simulations, measurements from a full OMPS-LP orbit are simulated using a two-dimensional ozone field. The resulting radiances are then used in both the one and two dimensional retrievals. To isolate the effects of horizonal ozone gradients, the input aerosol and albedo fields are assumed to be known and horizontally homogenous.

Figure 8 shows the results of the simulated retrieval for OMPS-LP orbit 20567. Qualitatively there is good agreement

between the one and two dimensional retrievals and the true ozone field, providing confidence in both methods. The two-dimensional retrieval agrees to better than 5% with the true ozone profile almost everywhere, with a few exceptions below $20\,\mathrm{km}$. Looking at the $15.5\,\mathrm{km}$ slice of the retrieval (top right panel of Fig. 8) it can be seen near $50°\,\mathrm{S}$ that the two-dimensional retrieval smooths out some of the fine oscillatory structure of the true profile, which is expected from the form of the averaging kernel. That being said, the two-dimensional retrieval captures the large ozone gradient in the $75°\,\mathrm{S}$ to $60°$ region very well.

Overestimation by the one-dimensional retrieval can be seen in the $75°\,\mathrm{S}$ to $60°$ region, $10$–$20\,\mathrm{km}$ region. The $15.5\,\mathrm{km}$ slice reveals that the one-dimensional retrieval assigns the horizontal gradient to the wrong location, leading the true profile by $\sim 2°$. Consistent overestimation by the one-dimensional retrieval is what would be expected by the measurement geometry and input ozone field. As OMPS-LP looks backward in the orbital plane, measurements near the edge of the polar vortex consistently look through high ozone values into lower ozone values. For limb scatter measurements ozone sensitivity is larger on the

instrument side of the line of sight (for an indepth discussion of this effect see Zawada et al., 2017), the high ozone values near OMPS-LP are incorrectly assigned to tangent points inside or near the vortex.

### 5.2    Example retrieved orbit with OMPS-LP

Figure 9 shows the retrieved ozone number density for OMPS-LP orbit 27695. Several low ozone filaments above the ozone layer are visible in both the southern hemisphere and northern hemisphere tropics/mid-latitudes. In the northern hemisphere a

low pocket of ozone can be seen below and intruding into the ozone layer.

Processing of this orbit took approximately 124 minutes using 8 threads on an i7-4770k cpu. There were 159 vertical images of radiance data input to the retrieval, giving an approximate processing time of 47 seconds per vertical image. Thus performing



**Figure 8.** Simulated retrieval results for OMPS-LP orbit 20657 near the polar vortex. The left column shows the true ozone field (top), tomographically retrieved ozone (middle), and the one-dimensionally retrieved ozone (bottom). The right column contains a horizontal slice of the retrieved ozone at 15.5 km (top), the percent difference between the tomographic retrieval and the truth (middle), and the percent difference between the one-dimensionally retrieved ozone and the truth (bottom). For the percent difference panels contours are shown every ±5%.





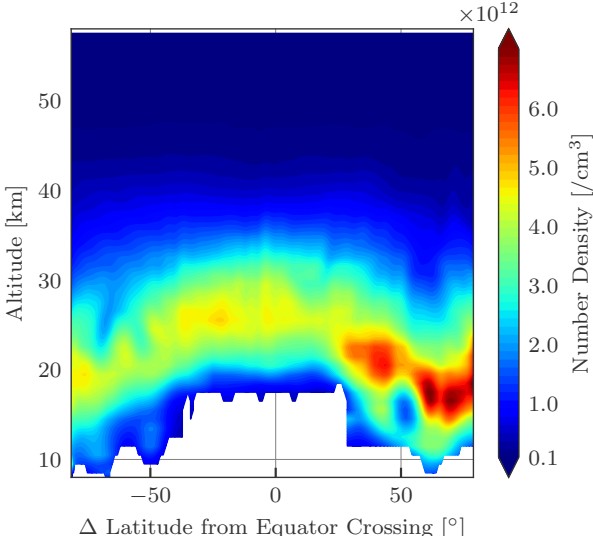

**Figure 9.** Retrieved ozone number density for OMPS-LP orbit 27695 (March 2nd, 2017, 10:30 AM UTC at equator crossing).

the 2D retrieval is not onerous from a computational point of view, two machines of similar computational power are sufficient to keep up to date with routine processing.

### 5.3 Monthly zonal mean anomalies

As a zeroth order valdiation effort monthly zonal mean relative ozone anomalies have been performed against the MLS v4.2
ozone measurements. The MLS retrievals (Livesey et al., 2006) native product is volume mixing ratio on pressure surfaces, for these comparisons we have converted MLS v4.2 measurements to number density on altitude levels using ERA-interim reanalysis (Dee et al., 2011). Figure 10 shows the result of these comparisons in the tropical $5°$ S to $5°$ N latitude bin. Qualitatively there is excellent agreement, the anomalous change in the QBO beginning at the end of 2015 can clearly be seen in both datasets. Quantitatively, above $25\,\mathrm{km}$ observed differences in relative anomaly are less than 0.05 ($\sim 5\%$ change in ozone)
for all time periods. Below $25\,\mathrm{km}$ differences on the order of 0.1 are seen, which could be related to larger variability in the tropical UTLS.

### 5.4 Near perfect coincidences with MLS

MLS onboard Aura and OMPS-LP onboard Suomi-NPP are both in sun-synchronous orbit with similar inclination and local crossing times, however Suomi-NPP orbits near $\sim 800\,\mathrm{km}$ while Aura is at $\sim 700\,\mathrm{km}$. The slight difference in orbital periods
causes the measurement ground tracks to drift relative to each other, with near perfect overlap, in both space and time, every 2–3 days. Figure 11 shows the measurement track of OMPS-LP orbit 11915, also shown are the available measurements from MLS which are nearly perfect coincident to the OMPS-LP measurements. At the crossing point there is a time difference of





**Figure 10.** Monthly zonal mean ozone anomalies in the $5°$ S to $5°$ N bin for OMPS-LP (top), MLS v4.2 (middle), and their absolute difference (bottom). Anomalies are calculated relative to the common overlap period, and data is masked outside the common overlap period.





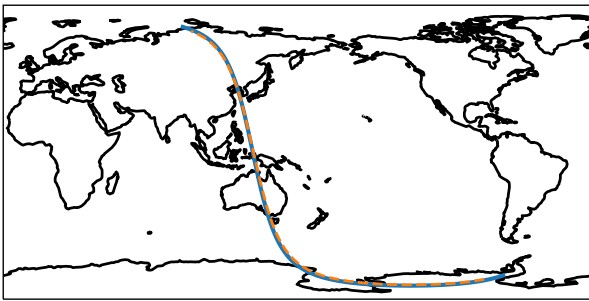

**Figure 11.** An example of near perfect coincident measurements from OMPS-LP and MLS. The dashed orange line shows the retrieval grid points for OMPS-LP orbit 11915, while the blue line shows the retrieval locations for the near coincident MLS measurements. The time difference at the crossing point is ∼16 minutes.

16 minutes, and the differences in longitude are less than $1°$ for the entire orbit track. It should be mentioned that sampling differences in latitude do not play a large factor as both the MLS and OMPS-LP retrievals are two-dimensional with the horizontal along-track resolution being poorer than the sampling frequency.

    The MLS retrieval is also two-dimensional, and has similar along-track resolution to the two-dimensional OMPS-LP retrie-
val, thus we have not applied horizontal averaging kernels for these tests. To account for differences in vertical resolution the OMPS-LP data has been degraded to the MLS pressure grid with a least squares fit, and then converted back to the altitude grid in a consistent fashion, however internal tests have shown that this makes negligable differences. A full validation of the dataset is intended for a forthcoming publication, however an initial validation check has been performed by examing near coincident orbits between Aura (MLS) and Suomi-NPP (OMPS-LP).

Figure 12 shows the retrieved ozone for OMPS-LP orbit 11915 and coincident MLS measurements. Qualitatively there is excellent agreement between the two retrievals. A triple ozone peak at low altitudes is seen in both retrievals in the northern hemisphere, and both retrievals resolve a break in the ozone peak near $40°S$. Some slight horizontal oscillations are observed ($\sim \pm 5\%$) in the USask OMPS-LP retrieval near the equator. The exact cause of the oscillations is currently unknown, but initial investigation suggests that it could be caused by the combination of cloud cover affecting the large amount of upwelling
observed due to low solar zenith angles ($\sim 20°$) seen in the tropics.

    Quantitatively agreement between the two retrievals (bottom panel of Fig. 12) is better than $5\%$ for the majority of the stratosphere. Differences greater than $10\%$ are seen at the lowest altitudes of the retrieval grid, it is possible that these are caused by the non-linearity involved in applying the pointing correction to the retrieved profile rather than the measurement tangent altitudes themselves. At the northern edge of the retrieval grid there are also differences on the order of $5-10\%$
which could be indicative of an edge effect in the two-dimensional retrieval. Above $45\,\mathrm{km}$ there is a large amount of variance observed, however this is reflected in the MLS precision estimate ($\sim 20\%$ at $0.5\,\mathrm{hPa}$).

    Figure 13 compares the same orbit in three latitude bins, $80°$ S to $40°$ S, $20°$ S to $20°$ N, and $40°$ N to $80°$ N. Also shown in Fig. 13 with the observed standard deviation, providing confidence in the supplied precision values for both OMPS-LP





**Figure 12.** The top panel shows the retrieved ozone field for OMPS-LP orbit 11915 from the USask 2D v1.0.2 retrieval, and the middle panel shows the corresponding coincident MLS v4.2 retrieved values for the coincident measurements shown in Fig. 11. The bottom panel shows the percent difference between the two, with gray and black contours indicating the $\pm 5\%$ and $\pm 10\%$ levels respectively.





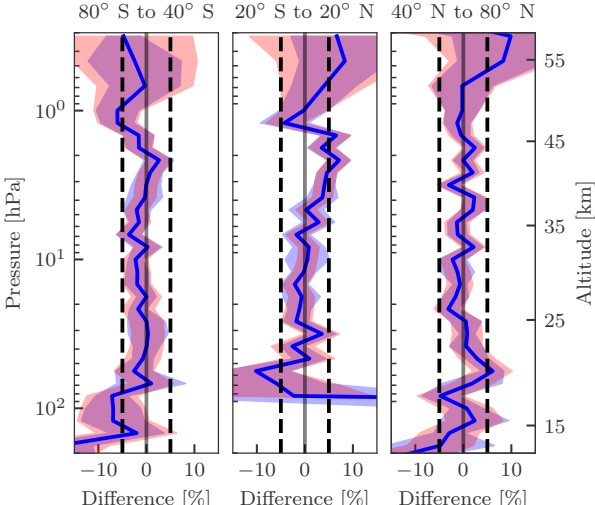

**Figure 13.** Mean differences ((OMPS-LP - MLS) / MLS · 100%) in three latitude bins for the comparison shown in Fig. 12. The shaded blue region shows the standard deviation of the differences, while the shaded red region is the predicted standard deviation using the precision estimate from both retrievals. Dashed vertical lines indicate the ±5% levels.

and MLS. However since each bin contains only roughly 30 measurements, these statistics should only be viewed as a rough qualitative estimate. In-depth studies using a large number of coincident orbits are planned for future validation efforts.

Lastly, results are shown for OMPS-LP orbit 20657 (October 23rd, 2015) are shown which are also nearly perfectly coincident to measurements from MLS. For this orbit we also apply the one-dimensional retrieval described in Sec. 5.1, and the
retrieval results are shown in Fig. 14. Similar to the previous orbit, agreement in the middle stratosphere is typically better than 5% between MLS and the two-dimensional retrieval. The one-dimensional retrieval also agrees favorably with MLS in the middle stratosphere. Inside the vortex there is larger disagreement, however this is expected due to the low absolute ozone values and the inherent variance of the retrievals.

Highlighted in Fig. 14 (dashed lines) is the 60° S to 75° S, 10–20 km region, which is the region where the one-dimensional
retrieval performed poorly in the simulations of Sec. 5.1. In this region the two-dimensional retrieval agrees better with MLS than the one-dimensional retrieval, with the one-dimensional retrieval consistently overestimating the ozone values. The 15.5 km slice (top right panel of Fig. 14) shows the one-dimensional retrieval leading both MLS and the two-dimensional retrieval, which is consistent with the prior simulation results. If we interpret the difference between the profiles at 15.5 km entirely as a latitudinal offset then the difference between the two-dimensional retrieval and MLS is ∼0.5° while the difference
between the one-dimensional retrieval and MLS is ∼2° at 65° S.



**Figure 14.** Retrieval results for OMPS-LP orbit 20657 near the polar vortex. The left column shows the coincident MLS v4.2 ozone (top), tomographically retrieved ozone (middle), and the one-dimensionally retrieved ozone (bottom). The right column contains a horizontal slice of the retrieved ozone at $15.5\,\mathrm{km}$ (top), the percent difference between the tomographic retrieval and MLS (middle), and the percent difference between the one-dimensionally retrieved ozone and MLS (bottom). For the percent difference panels contours are shown every $\pm 5\%$. The dashed black box indicates the area in which the two-dimensional retrieval is expected to show improvement based upon the simulations of Sec. 5.1.



# 6 Conclusions

A two-dimensional retrieval algorithm which directly accounts for atmospheric variations in the along orbital track dimension has been developed for use with limb scatter measurements from OMPS-LP. The retrieval algorithm combines all measurements from the sunlit portion of the orbit and simultaneously fits the full ozone profile for the entire orbit. The vertical resolution of the retrieved profiles is estimated to be 1–2 km while the along-track resolution is controlled with a Tikhonov type second squared difference constraint, and is typically 300–400 km for retrievals from OMPS-LP. The estimated precision of the retrieved ozone product is 2–5% for the middle and upper stratosphere, with values increasing to 30% just above the tropopause. Simulated retrievals were shown indicating that the retrieval is working as expected, and offers improvement over traditional one-dimensional retrievals in areas of large horizontal gradients.

The retrieval algorithm has been applied to all measurements from the center slit of OMPS-LP from early 2012 to present to create a multi-year near-global ozone time-series. Tropical ozone anomalies from the dataset agree well with those from MLS v4.2, with differences greater than 5% of the ozone mean value only observed below 25 km.

A preliminary validation effort is presented comparing one orbit of measurements to coincident measurements from MLS. These measurements are near perfectly coincident with time differences of less than 20 minutes and longitude differences of less than $1°$. For the majority of the stratosphere differences are less than 5%, larger differences are seen at the edges of the retrieval grid. Qualitatively the precision estimate matches the observed scatter seen in the differences. Coincident comparisons during the 2015 ozone hole indicates that the two-dimensional retrieval and MLS agree qualitatively well at the edge of the polar vortex, whereas a tradiational one-dimensional retrieval is shown to systematically overestimate in this area.

# 7 Data availability

The USask OMPS-LP v1.0.2 2D dataset is available in HARMOZ format (Sofieva et al., 2013) from the Odin-OSIRIS FTP server (see http://odin-osiris.usask.ca/?q=node/280).

*Competing interests.* The authors declare that they have no conflict of interest.

*Acknowledgements.* This work was partially funded by the Canadian Space Agency, the Natural Sciences and Engineering Research Council of Canada, Science Systems and Applications, Inc., and the National Aeronautics and Space Administration's Goddard Space Flight Center. We would also like to thank the OMPS-LP team for assistance and for providing a high quality L1 data product.



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
