# Peer review of "Tomographic retrievals of ozone with the OMPS Limb Profiler: algorithm description and preliminary results"

_Atmospheric Measurement Techniques, 2017_

## Referee Comment (RC1) · Anonymous Referee #2 · 28 Sep 2017

Authors introduce 2D tomographic retrieval algorithm for ozone retrieval from scattered light limb measurements of OMPS. Although scattered light tomography for limb measurements has been demonstrated earlier, the concept is for the first time applied for ozone retrieval for real measurements of OMPS. I encourage publishing the manuscript once the following points are sufficiently addressed.

General comments:

- in the first algorithm description sections authors provide rather theoretical description of the inversion method without stating what physical quantities are used as measurement and what exactly are going to be retrieved. Only very late and in different places

it is said that state vector may consist of various quantities in a sequence, e.g. logarithm of number density (stated only on page 13 of the manuscript), perhaps aerosol number density (but it is not clearly stated in Sect. 2.7.2). This makes it hardly possible to follow the arguments about practical considerations provided along the theoretical descriptions. I would strongly encourage the authors to restructure the manuscript to make its understanding straight forward.

- I have a feeling that more should be done with respect to the verification/validation, especially under strong gradient conditions. There is only one such orbit provided. Please add a study for a northern winter day with strong northern polar ozone depletion. For this case the Sun's geometry is opposite to that of the gradient in the SH. Also gradient at high SZAs (see below) must be investigate to sustain statements in the manuscript. Additionally middle and high latitudes where much stronger ozone variations might take place as at equator must be covered in a more systematic way. One could at least provide comparisons for one orbit per season thus covering typical seasonal variations in ozone distribution.

-there is a constant signal to noise ratio 100 assumed for the whole scan profile for the error estimation as given in conclusions of Jaross et al., 2014. Some sceptics is there due the natural illumination changes of several magnitudes along the tangent altitude and even despite the applied dynamical considerations, stray light and possible degradation of the instrument might be an issue.

Detailed comments:

P1L6: Add some words that MLS measurements used for the comparison are as well 2D, tomographic.

P1L24 add "and OClO" after "of NO2" since Pukite et al., 2008 did 2D retrieval for this gas as well.

P2L3 "relatively fast along orbital track sampling": fast relates to speed or time, perhaps

say "relatively fine resolved". Sect.2. As said in general comments; it would help a lot to state at the beginning what physical quantities you are operating with.

P3L7 "Between grid points bi-linear interpolation is applied to create a continuous representation of the atmosphere." It must be explained in detail how the interpolation is implemented. I.e. this could mean some subgridding or analytic constrains in model.

P3L15, F1 Figure must be improved. Please use different colors as "white grey" and other grey since it is really impossible to distinguish in the figure.

P3L17 "A common approach to minimize" Citations needed

P4L1-2 "Under this approach we have not noticed unphysical effects at the edges of the retrieval." A prove for this statement is necessary. Given your verification and validation evidence (just one orbit with gradient at lower SZA) this has not been verified: Can this been tested with an example with a gradient condition at the orbit parts with SZA 88 deg and above? In such cases Pukite et al. 2008 reported problems for the first profile of the orbit. Please provide evidence.

P4L10 Related to general comments. Still on the 4th page of the manuscript there is no idea what is to be state vector and measurement vector.

P4L18 A more concrete and exact description is needed. How the transformation is practically performed? What assumptions used? What has to be understood under "atmosphere specified ... is transformed", i.e. What is this atmosphere consisting from and characterized by? What and how it is changed due to transformation? How the Jacobian matrix is transformed?

P4L21 "These transformations are typically quite small in effect" Can you provide a number?

P5L6 And how much time resource do you need for one orbit?

P6L16 "Most atmospheric retrieval methods fall into two classes" Again, it is of course

good to give some review about the background of inverse methods but it is difficult for a reader to follow your considerations and choices if it is still not stated what you are going to retrieve from what.

P6L17 ""the resolution of the retrieved profile is determined by ... the resolution of the retrieval grid." This statement is generally wrong: The resolution is an ability to resolve some features. If there is not enough information one is not able to resolve the features even on fine grid. I think you wanted to say something else; perhaps one should skip the part of the sentence after "i.e." Eq.3 shouldn't all zeros be bold?

P7L16 "ozone number density, stratospheric aerosol number density, and surface reflectance assuming a Lambertian surface" Later you state that the state vector for ozone retrieval is logarithm of number density. This is again the confusion here between the long theory description and rather imprecise and misplaced description of the practical stuff.

P7L19 Does it mean solving 3 different separate inverse problems (Eq. 2)?

P8L20-21, Eq4 In text you mention k to be used for both indexing tangent altitude and triplet, though in Eq. (4) indexing for triplets is missing.

P8L22 What is meant by ozone sensitivity is minimal? Or perhaps effect of ozone absorption on spectra is minimal?

P13L8 "signal to noise ratio of 100"; "an upper bound on the error estimate taken from Jaross et al. (2014)." As said at the beginning this assumption might be much too optimistic.

P13L9 "state vector is the logarithm of number density". Only on page 13 there is finally mentioned the physical quantity all about the theory was. What about other quantities?

P14 Have you studied the effect different settings of the horizontal regularization. Is it not possible to do retrieval without any horizontal regularization because you also match the horizontal retrieval grid to that of the measurements?

P15L28 "orbit 27695" mention here day and time of the Eq. crossing.

P17L7 "Figure 10 shows the result of these comparisons in the tropical 5◦ S to 5◦ N latitude bin." What is about systematic study for other latitudes where far more gradients appear?

P19L10 day, time for orbit?

P23L4 "for the entire orbit" The retrieval is limited to SZA 88 deg. This should be stated.

P23L13 "one orbit" You compared two orbits.

P23L18 "tradiational"-> "traditional"

---

## Referee Comment (RC2) · A. Rozanov (Referee) · 13 Oct 2017

**Referee report to "Tomographic retrievals of ozone with the OMPS Limb Profiler: algorithm description and preliminary results" by D. Zawada et al.**

The manuscript presents a new 2-D algorithm to retrieve vertical distributions of ozone from OMPS-LP measurements. At vortex edge conditions with high meridional gradients of ozone, the algorithm is shown to perform better than a commonly used 1D retrieval. This fact makes the paper highly important for the scientific community. From the scientific and technical point of view the manuscript exhibit, however, several significant shortcomings related to the quality of the algorithm description, validity of the

approach and the used data, and the extention of the validation and analysis. A detailed list of the issues is provided in my general comments below. To my opinion the manuscript will be suitable for publishing in AMT after a major revision considering all my comments.

**General comments**

- Authors use outdated versions of OMPS Level 1 data (v2.0-2.4) although the new data version v2.5 is available already since May 2017. As version 2.5 already includes the pointing correction described in Sect. 3 of the manuscript this section would not be necessary any more if new Level 1 data was used.

- Pointing accuracy is mentioned as the main error source and the corrections in the order of 200-300 m seem to be considered by authors as important, otherwise one would rather skip Sect. 3. On the other hand, the authors do not hesitate to neglect the field of view of 1.5 km without making any considerations about the impact of this decision. As the field of view illumination is vertically inhomogeneous, I assume the neglect of field of view integration should have a similar effect as a misspointing. In this regard it is not quite clear why a very good agreement with MLS is still achieved and if the entire verification results might be accepted as trustable. To my opinion the evaluation must be repeated taking into account the field of view of the instrument.

- As an improvement of the retrieval quality by using a 2D retrieval is a key topic of the manuscript, synthetic retrievals as done in Sect. 5.1 need to be presented for the whole orbit. This is necessary to assess if smoothing out the small latitudinal variations by 2D retrieval as seen around 50°S in Fig. 8 is a general drawback of this approach or just an insignificant outlier. Furthermore, a similar study should be performed for another season with a vortex edge in the northern hemisphere. This will allow the reader to assess how the viewing geometry affects the relative performance of the 1D and 2D retrievals. Another important question is how the

retrieval results depend on the ozone distribution used to initialize the radiative transfer model. This question has not been addressed in the manuscript at all.

- The retrieval description is too much general with a lot of details hidden from the reader. For example, no or only insufficient quantitative information is provided about the latitudinal grid, reference tangent height and regularization parameters ($\gamma$ in Eq. (2) and $\alpha$ in Eq. (3)). The authors state that the a priori state vector is set to zero but make no comments about the values used to initialize the radiative transfer model. Are they also zero at the first iteration? The valid altitude range of the retrieval in not clearly identified.

- The validation is not sufficient to demonstrate the overall performance of the algorithm. The monthly mean comparison plots similar to Fig. 10 must be provided for absolute values rather than for anomalies for several latitude bands (tropics, middle and high latitudes).

**Detailed comments**

- Page 2, line 24: "... $\gamma_i I$ might be included..." - please make a clear statement if this term is included in your retrieval or not, if yes, what is the starting value and a typical end value of $\gamma_i$?

- Sect. 2.2: State vector is described insufficiently. Both altitude and latitude grids must be specified exactly providing the upper and lower limits as well as the sampling.

- Page 3, line 13: "A consequence of the limb viewing geometry..." - this is not a general consequence of the limb viewing geometry as a scanning instrument can be operated to avoid this problem (e.g. SCIAMACHY). This is rather a consequence of the imaging technique (2D detector array) used in OMPS.

- Page 3, paragraph starting at line 16: this is an unnecessary general discussion which do not provide any useful information. It is highly questionable if the method described by authors is really that general as no references are provided. Furthermore, possible griding issues vary with the observation method. For example the issues are completely different if a combination of measurements along and across the flying direction is used. I recommend to remove the paragraph and focus on the detailed description of the setup used in the retrieval rather than discussing any "general" approaches.

- Page 4, Sect. 2.3, starting from line 16 till the end of the section: to my opinion this text does not provide any useful information as for the retrieval/modeling description it is absolutely irrelevant whether the model performs the internal transformation of the coordinates or not. If you think it is important you need to describe it in much more details to give the reader understanding what is performed, how and for what reason, and which implications it can cause. Otherwise the text must be deleted as in its current form it is just confusing.

- Page 5, lines 1-2: "The sparsity of the Jacobian matrix can be improved..., as is done in Livesey et al. (2006)" - there are a lot of things which "can be done". The essential information is, however, if it "is done" in your retrieval or not. Please provide the numbers if it is done or clear statement that it is not done otherwise.

- Page 5, lines 3-6: This text does not contain any useful information. The matrices to be stored and inverted are already known from Eq. (2), their dimensions are already discussed in the first paragraph of the section, the fact if you solve the linear equation system using a solver for sparse or dense matrices is an absolutely minor technical information and a calculation of a memory space needed to store a $10000 \times 10000$ matrix is a very simple arithmetical exercise which is not relevant for a scientific paper.

- Page 5, last paragraph: the paragraph is quite confusing. It not strictly defined

what you understand as a "forward model run". In any case you have to simulate the radiance for every measured pixel, otherwise you just loose the information. Formally you can do just one "forward model run" and simulate everything. Thus, to understand this discussion, the reader has to know what is meant as a "run". Normally the forward model is run for each internal grid point, this might coincide with the location of the image or not. Surely a reduction of grid points reduces the computation time. So, actually, you just need to provide the information on the latitudinal grid and skip the remaining discussion.

• Page 6, line 1: "10° cone" - commonly the term "cone" is used for a 3D object while you have a 2D approach. Please use a proper notation. Furthermore, it is unclear how this "cone" is defined, I suppose from the Earth's center, but it should be clearly stated to avoid a confusion.

• Page 6, line 3: "Each image..." - do you mean that the solar zenith angle changes from image to image? It is actually obvious that the illumination and composition of the atmosphere changes from one location to anther. Why is it an issue?

• Page 6, line 4: "... internal atmosphere is specified as a plane" - I suppose you mean the meridional direction. It should be clearly stated to avoid a misinterpretation.

• Sect. 2.5: Actually I did not find anywhere a statement about the variable defining the along-orbit grid, is it latitude, solar zenith angle, of anything else?

• Sect. 2.5: The last paragraph does not contain any useful information as it is not discussed how the OSIRIS images are compiled and how the corresponding radiative transfer calculations are done. Surely the listed conditions are not an issue for 1D retrievals if each observation is processed independently. I recommend to remove the paragraph.

- Sect. 2.6: Remove the first two paragraphs of the section. These paragraphs pretend to provide an overview of the methods fail however to do that as the discussion is to sketchy. Furthermore, this information is not needed for the discussion below.

- Page 6, line 28: "For our retrieval ...": please bear in mind that $\gamma_i I$ also works as a regularization term. So, when using Levenberg-Marquardt approach it is incorrect to state that the retrieval is completely unregularized. By the way, it is still not clearly stated if you use the Levenberg-Marquardt term in you approach or not.

- Page 7, Eq. (3): Provide $\alpha$ value.

- Page 7, Eq. (3): The statement "**0** indicates a number of zeros equal to the number of altitude grid points" is wrong. It must be the number of altitude grid points minus one.

- Page 7, line 4: There are certainly some good reasons to use zero a priori state vector especially when employing smoothing constraints but the "simplicity" is not really the best one. It should be also mentioned that usage of zero a priori state vector often results in a low bias of the solution.

- Page 7, lines 9-10: I do not agree that the resolutions of the vertical and horizontal grids are strictly coupled. In principle any grid combinations can be used, this might require however a stronger regularization as the total amount of information remains obviously the same. The main challenge here is to identify the optimal set of grids and regularization parameters. This set might however depend on the targeted usage of the retrieval data.

- Page 7, lines 12-13: "The effect of a one dimensional retrieval on horizontal regularization...." - I guess you mean "horizontal resolution".

- Table 1: Please provide the reference tangent height for each interval.

- Sect. 2.7.1: What is the minimum retrieval altitude for ozone?

- Page 8, line 7: Here and further below in the text you are talking about the "atmospheric upwelling". I suppose you mean the upwelling radiation. However, this notation is commonly used in the scientific community to describe the dynamic processes and means the upward moving air masses rather than radiance. Please use another notation throughout the text to avoid a confusion.

- Page 8, lines 20-21: I guess Eq. (4) is valid for both triplets and doublets. "... for triplet $k$" in line 21 should be "... for triplet $l$".

- Page 9: "...any errors in the absolute calibration ..." - this is not completely true for an imaging instrument because the information for different tangent heights comes from different areas of the CCD and can have different calibration errors.

- Page 10, Eq, (6): It is not clear how the second term is employed in the retrieval as the modeled Rayleigh background needs to be subtracted in the same way from both measured and modeled radiances and thus is canceled out when calculating $y - F(x)$ in accordance with Eq. (2).

- Sect. 2.7.2: No information is provided about how the aerosol extinction coefficient is calculated for other wavelengths.

- Page 11, line 1: "... albedo is handled in a two-dimensional sense ..." - what is the second dimension for the albedo?

- Sect. 2.7.3: 40 km tangent height to retrieve the surface albedo is quite high. Have you checked a possible influence of the stray light at this tangent height?

- Sect. 2.7.3: The influence of the albedo spectral dependence must be discussed. For example, for a green vegetation the albedo obtained at 745 nm can be very different from that at 602 nm (red edge).

- Sect. 3: The section is unnecessary as all discussed corrections are already implemented in the Level 1 v2.5 dataset of NASA.

- Sect. 4: If Levenberg-Marquardt term is used in the retrieval it must be also included in the error analysis.

- Sect. 4: Is the signal to noise of 100 is used only in the error analysis or in the standard retrieval as well? Why was not the signal to noise data provided in Level 1 data set used? The latter would provide a realistic instead of maximum error estimation.

- Page 13, line 9: Only in the error analysis section the reader learn that the logarithm of the number density is the retrieval parameter rather than the number density itself. This must have been mentioned already in Sect. 2.2.

- Page 13, line 14: what does "but near where the tropopause lowers at mid-latitudes" refer to?

- Fig. 7: Suboptimal color scale. How is the sign of the distance from the retrieval location defined?

- Page 14, line 3: "Since the regularization term..." - once again, do not exclude the Levenberg-Marquardt term from the discussion.

- Fig. 7: The definition of the vertically/horizontally integrated averaging kernels is not quite clear. You have a set of averaging kernels for each vertical/horizontal grid point and each of them spans in both vertical and horizontal directions. Is the integration done over these directions? Is yes you seem to show one averaging

kernel at each altitude in each panel in Fig. 7? If it was true I would expect the plural in the beginning of line 8 as you show multiple averaging kernels for different altitudes in each panel of Fig. 7. If my understanding of the definition is correct, I would like you to explain why there is a clear maximum at 40 km in tropics and 45 km at mid-latitudes and how it can be interpreted in terms of the retrieval sensitivity.

- Page 14, line 9: "Only minor differences ..." - to my opinion the majority of differences occur around 40 km and they are not minor.

- Page 14, lines 11-12: "it was found that ..." - it is hard to believe as it is widely known that the averaging kernels for "relative" retrievals (i.e. retrieval of relative deviations from a priori or logarithms) depend on the atmospheric state. Please provide the averaging kernel plot for different season to justify you statement.

- Fig. 7: why does the tropics plot have a white area below 18 km, how is the lower boundary of the retrieval defined?

- Page 14, last paragraph: It is absolutely inappropriate to neglect the instrument field of view without any investigations as it might lead to a significant change in both the retrieval results and error analysis.

- Sect. 5.1: The results must be provided over the whole orbit as it is essential to estimate how the retrievals compare outside the vortex edge region. Another simulation for a different season with a vortex edge in the northern hemisphere needs to be provided to assess the influence of the viewing geometry.

- Page 15, line 25: "For limb scatter measurements ..." - please illustrate this by plotting the averaging kernel for about 65°S and 15.5 km in both horizontal and vertical directions using a proper color scale.

- Sect. 5.2: this section is not really informative and can be skipped. Details on the execution time suit better in the algorithm description section.

- Sect. 5.3: Not only the anomalies but also the monthly mean values themselves need to be compared. This needs to be done for different latitude bands (tropics, mid-latitudes, high latitudes).

- Fig. 10: Why the altitudes above 59 km are not shown? If I understand it correctly, the retrieval runs up to 59 km.

- Page 19, lines 1-3: "... with the horizontal along-track resolution being poorer.." - please provide the values of the resolution and sampling for both instruments.

- Page 19, lines 5-6: ".. has been degraded to the MLS pressure grid with a least square fit..." - please clarify what exactly was fitted and how you can degrade the vertical resolution using a least square fit. Here, a convolution with averaging kernels would be more suitable.

- Fig. 12: Provide the lower and upper altitude of the plotted range. Provide the same plot from 1D retrieval. Explain the lower limit of the retrieval.

- It would be also interesting to show some examples from NASA Level 2 data, especially in Fig. 14.

**Technical corrections**

- Page 2, Eq. (1) matrices have to be shown in bold face to match the corresponding notations in the text.

- Page 15, line 5: duplicated word "those"

---

## Author Comment (AC1) · 24 Nov 2017

We would like to thank the referee for their helpful comments and suggestions. Included below is each of the referee's comments (italics) followed by our reply.

**Responses to Anonymous Referee 2**

**General Comments**

*in the first algorithm description sections authors provide rather theoretical description of the inversion method without stating what physical quantities are used as measurement and what exactly are going to be retrieved. Only very late and in different places it is said that state vector may consist of various quantities in a sequence, e.g. logarithm of number density (stated only on page 13 of the manuscript), perhaps aerosol number density (but it is not clearly stated in Sect. 2.7.2). This makes it hardly possible to follow the arguments about practical considerations provided along the theoretical descriptions. I would strongly encourage the authors to restructure the manuscript to make its understanding straight forward.*

**Reply**: Thank you for the suggestion, we agree in many places there was too much general information. In the revised manuscript specifics about the state vector (such as that it is logarithm of number density) appear much earlier. Some of the general information has been removed and/or replaced with specifics about the retrieval for OMPS-LP. We have also rearranged some sections so that information about the specific species retrievals (ozone, aerosol, albedo) appears earlier.

*I have a feeling that more should be done with respect to the verification/validation, especially under strong gradient conditions. There is only one such orbit provided. Please add a study for a northern winter day with strong northern polar ozone deple- tion. For this case the Suns geometry is opposite to that of the gradient in the SH. Also gradient at high SZAs (see below) must be investigate to sustain statements in the manuscript. Additionally middle and high latitudes where much stronger ozone variations might take place as at equator must be covered in a more systematic way. One could at least provide comparisons for one orbit per season thus covering typical seasonal variations in ozone distribution.*

**Reply**: We have added a second simulated orbit where there is strong ozone depletion in the northern hemisphere. We agree that there is a lot of potential validation work left to do for the dataset but we believe it is beyond the scope of the manuscript. As stated in the text, we are planning on doing doing an in-depth validation in a future study, what is presented here is preliminary efforts in this direction. We have replied to the referees comment concerning gradients at high solar zenith angles in the specific comments below.

*there is a constant signal to noise ratio 100 assumed for the whole scan profile for the*

*error estimation as given in conclusions of Jaross et al., 2014. Some sceptics is there due the natural illumination changes of several magnitudes along the tangent altitude and even despite the applied dynamical considerations, stray light and possible degradation of the instrument might be an issue.*

**Reply**: This could certainly be the case, but currently the estimate given in Jaross et al., 2014 remains the best available estimate for the instrument performance. The effect of illumination changing by several orders of magnitude is somewhat mitigated by the OMPS-LP instrument which collects two interleaved images of the full scene with different integration times and aperture sizes. It does appear that the final error analysis results are however at least somewhat reasonable from the standard deviation of the MLS comparisons.

**Specific Comments**

**P1L6** *Add some words that MLS measurements used for the comparison are as well 2D, tomographic.*

**Reply**: Added

**P1L24** *add "and OClO" after "of NO2" since Pukite et al., 2008 did 2D retrieval for this gas as well.*

**Reply**: Added

**P2L3** *"relatively fast along orbital track sampling": fast relates to speed or time, perhaps say "relatively fine resolved".*

**Reply**: Changed

**Sect.2.** *As said in general comments; it would help a lot to state at the beginning what physical quantities you are operating with.*

**Reply**: This is done in the revised manuscript.

**P3L7** *"Between grid points bi-linear interpolation is applied to create a continuous representation of the atmosphere." It must be explained in detail how the interpolation is implemented. I.e. this could mean some subgridding or analytic constrains in model.*

**Reply**: This is a good point. We have moved this statement to the forward model section

to make it clear this is something internal to the forward model. The exact procedure in which this is done is described in detail in Zawada et al. 2015.
* * *
**P3L15, F1** *Figure must be improved. Please use different colors as "white grey" and other grey since it is really impossible to distinguish in the figure.*

**Reply**: We have changed the two colors to red and gray.
* * *
**P3L17** *"A common approach to minimize" Citations needed*

**Reply**: We have added a reference to Rodgers (2000).
* * *
**P4L1-2** *"Under this approach we have not noticed unphysical effects at the edges of the retrieval." A prove for this statement is necessary. Given your verification and validation evidence (just one orbit with gradient at lower SZA) this has not been verified: Can this been tested with an example with a gradient condition at the orbit parts with SZA 88 deg and above? In such cases Pukite et al. 2008 reported problems for the first profile of the orbit. Please provide evidence.*

**Reply**: In the shown orbit there are natural gradients in the ozone field at gradients above 88 SZA, we have not artificially suppressed them. We have extended this orbit to the full range (rather than just the southern hemisphere) as well to see both edges of the retrieval. We have also added a second simulated orbit with gradient conditions in the northern hemisphere, and in none of these cases have we noticed significant edge effects. However it is true that if there was a large, ozone hole in magnitude, gradient occurring at 88 SZA (typically does not happen with the OMPS-LP geometry) we would expect some edge effects to occur. We have changed the offending statement to read "Under this approach for typical conditions we have not noticed unphysical effects at the edges of the retrieval, but this is still under investigation."
* * *
**P4L10** *Related to general comments. Still on the 4th page of the manuscript there is no idea what is to be state vector and measurement vector.*

**Reply**: This should be better in the revised manuscript, see reply to the general comment.
* * *
**P4L18** *A more concrete and exact description is needed. How the transformation is practically performed? What assumptions used? What has to be understood under "atmosphere specified on the retrieval grid is transformed", i.e. What is this atmosphere consisting from and characterized by? What and how it is changed due to transformation? How the Jacobian*

*matrix is transformed?*

**Reply**: This section was poorly worded and has mostly been removed at the request of the other referee. All that is done is a linear change of coordinates from the plane containing the lines of sight to the orbital track (retrieval grid). We have moved the information about the mismatch between the line of sight plane and the retrieval grid to the state vector section.
* * *
**P4L21** *"These transformations are typically quite small in effect" Can you provide a number?*

**Reply**: We have added the statement that at the equator the mismatch between the line of sight plane and the orbital plane is approximately $5°$.
* * *
**P5L6** *And how much time resource do you need for one orbit?*

**Reply**: This information was stated later in the manuscript. In the revised manuscript we have moved the example retrieved orbit section (which stated the approximate time per orbit) into the algorithm section.
* * *
**P6L16** *"Most atmospheric retrieval methods fall into two classes" Again, it is of course good to give some review about the background of inverse methods but it is difficult for a reader to follow your considerations and choices if it is still not stated what you are going to retrieve from what.*

**Reply**: This section has been removed at the request of the other referee.
* * *
**P6L17** *"the resolution of the retrieved profile is determined by ... the resolution of the retrieval grid." This statement is generally wrong: The resolution is an ability to resolve some features. If there is not enough information one is not able to resolve the features even on fine grid. I think you wanted to say something else; perhaps one should skip the part of the sentence after "i.e."*

**Reply**: At the request of the other referee we have removed this section so this is not an issue anymore. However, if we define resolution from the averaging kernel then as long as $K^T S_\epsilon^{-1} K$ is invertible then this is a true statement. So the referee is correct in that the grid can not be made arbitrarily fine, but if $K^T S_\epsilon^{-1} K$ is singular then the retrieval cannot be done without regularization anyways.
* * *
**Eq.3** *shouldnt all zeros be bold?*

**Reply**: We think it is correct the way it is. The non-bold zeros are needed to separate

different altitudes.
* * *
**P7L16** *"ozone number density, stratospheric aerosol number density, and surface reflectance assuming a Lambertian surface" Later you state that the state vector for ozone retrieval is logarithm of number density. This is again the confusion here between the long theory description and rather imprecise and misplaced description of the practical stuff.*

**Reply**: This should be improved in the revised manuscript, these statements have been moved earlier with much of the general information changed to OMPS-LP specific information.
* * *
**P7L19** *Does it mean solving 3 different separate inverse problems (Eq. 2)?*

**Reply**: Yes for three separate inverse problems, but the albedo retrieval does not use Eq. 2.
* * *
**P8L20-21, Eq4** *In text you mention k to be used for both indexing tangent altitude and triplet, though in Eq. (4) indexing for triplets is missing.*

**Reply**: Thank you, this has been corrected.
* * *
**P8L22** *What is meant by ozone sensitivity is minimal? Or perhaps effect of ozone absorption on spectra is minimal?*

**Reply**: We have reworded this to "where the effect of ozone absorption on the observed radiance is minimal."
* * *
**P13L8** *"signal to noise ratio of 100"; "an upper bound on the error estimate taken from Jaross et al. (2014)." As said at the beginning this assumption might be much too optimistic.*

**Reply**: See the reply to the general comment
* * *
**P13L9** *"state vector is the logarithm of number density". Only on page 13 there is finally mentioned the physical quantity all about the theory was. What about other quantities?*

**Reply**: Description of the state vector has been improved, this information appears much earlier in the revised manuscript.
* * *
**P14** *Have you studied the effect different settings of the horizontal regularization. Is it not*

*possible to do retrieval without any horizontal regularization because you also match the horizontal retrieval grid to that of the measurements?*

**Reply**: We have been looking into this. It is possible to do the retrieval without regularization, however there end up being unphysical oscillations in the horizontal direction. The exact cause of the oscillations is still being investigated but it would not be unexpected for a retrieval where the sampling matches the retrieval grid to have oscillations in that dimension. We chose the current level of regularization somewhat conservatively to remove the oscillations, and we plan to further study this for a future version of the retrieval.
* * *
**P15L28** *"orbit 27695" mention here day and time of the Eq. crossing.*

**Reply**: Added
* * *
**P17L7** *"Figure 10 shows the result of these comparisons in the tropical 5° S to 5° N latitude bin." What is about systematic study for other latitudes where far more gradients appear?*

**Reply**: A full systematic validation of the dataset is intended for a future validation paper.
* * *
**P19L10** *day, time for orbit?*

**Reply**: Added
* * *
**P23L4** *"for the entire orbit" The retrieval is limited to SZA 88 deg. This should be stated.*

**Reply**: Thank you, we have added this.
* * *
**P23L13** *"one orbit" You compared two orbits.*

**Reply**: Thank you, changed.
* * *
**P23L18** *"tradiational"-¿ "traditional"*

**Reply**: Thank you, corrected

---

## Author Comment (AC2) · 24 Nov 2017

We would like to thank the referee for their helpful comments and suggestions. Included below is each of the referee's comments (italics) followed by our reply.

**Responses to Referee 1 (Alexei Rozanov)**

**General Comments**

*Authors use outdated versions of OMPS Level 1 data (v2.0-2.4) although the new data version v2.5 is available already since May 2017. As version 2.5 already includes the pointing correction described in Sect. 3 of the manuscript this section would not be necessary any more if new Level 1 data was used.*

**Reply**: It is true that a similar pointing correction has been included in the v2.5 L1G product, we have mentioned this in the revised manuscript. However we feel it is important to include the section as the manuscript serves as a description of our v1.0.2 retrieved ozone product which has already been used in several studies. We are planning on producing a new version of our ozone data product in the future based on the new L1G product but it is not feasible to include it here due to the computational burden of reprocessing the entire mission.
* * *
*Pointing accuracy is mentioned as the main error source and the corrections in the order of 200-300 m seem to be considered by authors as important, oth- erwise one would rather skip Sect. 3. On the other hand, the authors do not hesitate to neglect the field of view of 1.5 km without making any considerations about the impact of this decision. As the field of view illumination is vertically inhomogeneous, I assume the neglect of field of view integration should have a similar effect as a misspointing. In this regard it is not quite clear why a very good agreement with MLS is still achieved and if the entire verification results might be accepted as trustable. To my opinion the evaluation must be repeated taking into account the field of view of the instrument.*

**Reply**: We agree that neglecting the field of view has the potential to have an effect on the retrieval, however this is no trivial matter. The referee states that the field of view is 1.5 km, which is approximately true for a single pixel on the detector, but the provided level 1 product is gridded, bilinearly interpolated from four neighboring pixels. The actual field of view (both magnitude and shape) varies as a function of altitude and wavelength depending on where each pixel is, and this information is not publicly available. Furthermore, neglecting the instrumental field of view is a common assumption in many limb retrievals (including the operational NASA OMPS-LP retrieval and the OSIRIS retrieval, which this work builds upon) and we do not feel it is within the scope of this manuscript to perform a full study on this effect.

However, as stated in the manuscript we do intend to investigate this further in a future version of the retrieval. We expect that neglecting the field view has a vertical smoothing

effect, which is why we do not trust our predicted 1 km vertical resolution and instead estimate it as 1–2 km. Our preliminary tests have indicated that neglecting the field of view has the potential to introduce a 1–2 % high bias near 20–25 km. This is considerably less than the ~7% error you could see with a 300 m pointing shift at high altitudes.
* * *
*As an improvement of the retrieval quality by using a 2D retrieval is a key topic of the manuscript, synthetic retrievals as done in Sect. 5.1 need to be presented for the whole orbit. This is necessary to assess if smoothing out the small latitudinal variations by 2D retrieval as seen around 50° S in Fig. 8 is a general drawback of this approach or just an insignificant outlier. Furthermore, a similar study should be performed for another season with a vortex edge in the northern hemisphere. This will allow the reader to assess how the viewing geometry affects the relative performance of the 1D and 2D retrievals. Another important question is how the retrieval results depend on the ozone distribution used to initialize the radiative transfer model. This question has not been addressed in the manuscript at all.*

**Reply**: We have changed the requested figure to show the entire orbit, and also added a second orbit using synthetic data with northern hemisphere ozone depletion. Internal tests have shown that the dependence on the profile used to initialize model is negligible ($\leq 1\%$), we have added a statement to the revised manuscript to this effect.
* * *
*The retrieval description is too much general with a lot of details hidden from the reader. For example, no or only insufficient quantitative information is provided about the latitudinal grid, reference tangent height and regularization parameters $\gamma$ in Eq. (2) and $\alpha$ in Eq. (3)). The authors state that the a priori state vector is set to zero but make no comments about the values used to initialize the radiative transfer model. Are they also zero at the first iteration? The valid altitude range of the retrieval in not clearly identified.*

**Reply**: We agree that the retrieval description was too general in many places. In the revised manuscript the description of aspects of the retrieval such as the state vector are given directly for the retrieval as applied to OMPS-LP, rather than first for a theoretical retrieval. Many of these specific changes are outlined as replies to the referees other comments below.
* * *
*The validation is not sufficient to demonstrate the overall performance of the algorithm. The monthly mean comparison plots similar to Fig. 10 must be provided for absolute values rather than for anomalies for several latitude bands (tropics, middle and high latitudes).*

**Reply**: The provided comparisons are intended to demonstrate the validity of the technique and not be a full validation of the dataset, which we feel is beyond the scope of this manuscript. As stated in the manuscript the validation work presented is preliminary, and a full validation is planned for a future paper.

**Detailed comments**

**Page 2, line 24** *"... $\gamma_i I$ might be included..."* - *please make a clear statement if this term is included in your retrieval or not, if yes, what is the starting value and a typical end value of $\gamma_i$?*

**Reply**: We have added the statement that a small $\gamma_i = 0.1\cdot$ the mean value of the diagonal of $\mathbf{K}_i^T \mathbf{S}_\epsilon^{-1} \mathbf{K}_i$ term is included to primarily aid with the stability of the inversion.
* * *
**Sect. 2.2** *State vector is described insufficiently. Both altitude and latitude grids must be specified exactly providing the upper and lower limits as well as the sampling.*

**Reply**: This section has been rewritten, we have opted to immediately explain the state vector for ozone here rather than a generic state vector. The horizontal grid is not spaced in latitude, rather it is in angle along the orbital plane of OMPS-LP, we hope this is now clear. Upper and lower limits and sampling have been noted for both the vertical and horizontal components in the revised section.
* * *
**Page 3, line 13** *"A consequence of the limb viewing geometry..."* - *this is not a general consequence of the limb viewing geometry as a scanning instrument can be operated to avoid this problem (e.g. SCIAMACHY). This is rather a con- sequence of the imaging technique (2D detector array) used in OMPS.*

**Reply**: We have changed the wording to "A consequence of the OMPS-LP viewing geometry"
* * *
**Page 3, paragraph starting at line 16** *this is an unnecessary general discussion which do not provide any useful information. It is highly questionable if the method described by authors is really that general as no references are provided. Furthermore, possible griding issues vary with the observation method. For example the issues are completely different if a combination of measurements along and across the flying direction is used. I recommend to remove the paragraph and focus on the detailed description of the setup used in the retrieval rather than discussing any "general" approaches.*

**Reply**: We have removed this information.
* * *
**Page 4, Sect. 2.3, starting from line 16 till the end of the section** *to my opinion this text does not provide any useful information as for the retrieval/modeling description it is absolutely irrelevant whether the model performs the internal transformation of the coordinates or not. If you think it is important you need to describe it in much more details to give the reader*

*understanding what is performed, how and for what reason, and which implications it can cause. Otherwise the text must be deleted as in its current form it is just confusing.*

**Reply**: We have moved the figure and the text mentioning the mismatch between the instrument line of sight and the retrieval grid to the state vector section. We have also removed the text about the coordinate transformations as requested, and now simply state that the line of sight plane is projected onto the retrieval grid.
* * *
**Page 5, lines 1-2** *"The sparsity of the Jacobian matrix can be improved..., as is done in Livesey et al. (2006)" - there are a lot of things which "can be done". The essential information is, however, if it "is done" in your retrieval or not. Please provide the numbers if it is done or clear statement that it is not done otherwise.*

**Reply**: We have added the statement "For our retrieval we limit each measurement to contribute to profiles within 10° of the tangent point.".
* * *
**Page 5, lines 3-6** *This text does not contain any useful information. The matrices to be stored and inverted are already known from Eq. (2), their dimensions are already discussed in the first paragraph of the section, the fact if you solve the linear equation system using a solver for sparse or dense matrices is an absolutely minor technical information and a calculation of a memory space needed to store a 10000 × 10000 matrix is a very simple arithmetical exercise which is not relevant for a scientific paper.*

**Reply**: We agree that obviously calculating the storage requirements for a matrix is simple, but we do not agree that the information should be removed. One of the limiting factors for grid spacing, number of measurements used, etc., for a tomographic retrieval is computational. Other papers describing tomographic techniques such as Livesey et. al. 2006 and Ungermann et al. 2010 include similar types of information.

Ungermann, J., Kaufmann, M., Hoffmann, L., Preusse, P., Oelhaf, H., Friedl-Vallon, F., and Riese, M.: Towards a 3-D tomographic retrieval for the air-borne limb-imager GLORIA, Atmos. Meas. Tech., 3, 1647-1665, https://doi.org/10.5194/amt-3-1647-2010, 2010.

Livesey, N. J., Van Snyder, W., Read, W. G., and Wagner, P. A. (2006). Retrieval algorithms for the EOS Microwave limb sounder (MLS). IEEE transactions on geoscience and remote sensing, 44(5), 1144-1155.
* * *
**Page 5, last paragraph** *the paragraph is quite confusing. It not strictly defined what you understand as a "forward model run". In any case you have to simulate the radiance for every measured pixel, otherwise you just loose the information. Formally you can do just one "forward model run" and simulate everything. Thus, to understand this discussion, the reader has to know what is meant as a "run". Normally the forward model is run for*

*each internal grid point, this might coincide with the location of the image or not. Surely a reduction of grid points reduces the computation time. So, actually, you just need to provide the information on the latitudinal grid and skip the remaining discussion.*

**Reply**: We realize that this section was confusing for those who are not familiar with SASKTRAN. The point we were trying to convey is that the expensive part of the radiative transfer calculation is calculating the multiple scatter source function, $J_{MS}$, which is a function of space, atmospheric state, and time. Once we have $J_{MS}$ the final line integrals take little effort. The potential for computational time saving here is that since the grid points are close to each other, we can calculate $J_{MS}$ in a spatial region that covers multiple grid points. The problem is that each "run" of SASKTRAN is for a single instant of time, so doing that is not strictly valid because each measurement obviously does not occur at the same instant in the time. So in this section we attempted to describe the potential issues with assuming that $\sim 5$ measurements occur at the same instant of time (roughly 100 s), it has nothing to do with changing the number of grid points (the largest issue is that the sun is assumed to be in the same location over this 100 s). We have rewritten the majority of this section to try to make this more clear.
* * *
**Page 6, line 1** *"10° cone"* - *commonly the term "cone" is used for a 3D object while you have a 2D approach. Please use a proper notation. Furthermore, it is unclear how this "cone" is defined, I suppose from the Earths center, but it should be clearly stated to avoid a confusion.*

**Reply**: We have added the statement that the cone's vertex is the Earth's center, however in this case we believe that cone is the correct term. While the retrieval is a 2D approach, as soon as the atmosphere is allowed to vary in a second dimension (other than SZA) the symmetry in the source function is broken and it becomes 5 dimensional (position, direction) rather than 4 dimensional (altitude, SZA, direction). The source function is solved within this three dimensional cone.
* * *
**Page 6, line 3** *"Each image..."* - *do you mean that the solar zenith angle changes from image to image? It is actually obvious that the illumination and composition of the atmosphere changes from one location to anther. Why is it an issue?*

**Reply**: We hope that this is clearer now that we have rewritten this section. The issue is that each measurement happens at a different time, a single SASKTRAN-HR calculation is one instant of time, so modelling multiple measurements with one SASKTRAN-HR calculation involves an assumption.
* * *
**Page 6, line 4** *"... internal atmosphere is specified as a plane"* - *I suppose you mean the*

*meridional direction. It should be clearly stated to avoid a misinterpretation.*

**Reply**: This has been changed to "as a plane in the along line of sight direction".
* * *
**Sect. 2.5** *Actually I did not find anywhere a statement about the variable defining the along-orbit grid, is it latitude, solar zenith angle, of anything else?*

**Reply**: The grid is the angle within the orbital plane, we hope that in the revised manuscript this is clear.
* * *
**Sect. 2.5** *The last paragraph does not contain any useful information as it is not discussed how the OSIRIS images are compiled and how the corresponding radiative transfer calculations are done. Surely the listed conditions are not an issue for 1D retrievals if each observation is processed independently. I recommend to remove the paragraph.*

**Reply**: We think the modified section and our previous answers has made this clear. The issues are very much the same for the OSIRIS 1D retrieval which assumes that each scan happens at one instant of time, rather than running a new radiative transfer calculation for each individual observation.
* * *
**Sect. 2.6** *Remove the first two paragraphs of the section. These paragraphs pretend to provide an overview of the methods fail however to do that as the discussion is to sketchy. Furthermore, this information is not needed for the discussion below.*

**Reply**: We have removed these paragraphs.
* * *
**Page 6, line 28** *"For our retrieval ...": please bear in mind that $\gamma_i I$ also works as a regularization term. So, when using Levenberg-Marquardt approach it is incorrect to state that the retrieval is completely unregularized. By the way, it is still not clearly stated if you use the Levenberg-Marquardt term in you approach or not.*

**Reply**: We do not believe this is correct in the standard use of the term "regularization". The Levenberg-Marquardt term does not appear in the cost function as would a traditional regularization term, and in theory, the retrieval should converge to the same solution (neglecting issues of multiple local minima) with or without the Levenberg-Marquardt term. It is true that the Levenberg-Marquardt term can have a regularization effect if the retrieval is

stopped before proper convergence, but that is not the case here.
* * *
**Page 7, Eq. (3)** *Provide $\alpha$ value.*

**Reply**: Added.
* * *
**Page 7, Eq. (3)** *The statement " **0** indicates a number of zeros equal to the number of altitude grid points" is wrong. It must be the number of altitude grid points minus one.*

**Reply**: Thank you, this has been corrected.
* * *
**Page 7, line 4** *There are certainly some good reasons to use zero a priori state vector especially when employing smoothing constraints but the "simplicity" is not really the best one. It should be also mentioned that usage of zero a priori state vector often results in a low bias of the solution.*

**Reply**: We have removed the word "simplicity". We agree that with certain forms of regularization a zero a priori results in a low bias, however we have not seen anything to suggest that a second derivative constraint results in a consistent low bias. If there is a study that shows this we would be happy to state this and add a reference.
* * *
**Page 7, lines 9-10** *I do not agree that the resolutions of the vertical and horizontal grids are strictly coupled. In principle any grid combinations can be used, this might require however a stronger regularization as the total amount of information remains the same. The main challenge here is to identify the optimal set of grids and regularization parameters. This set might however depend on the targeted usage of the retrieval data.*

**Reply**: I think we are mostly in agreement here, when we say the resolutions are coupled we meant to refer to the resolution of the retrieval, not the actual grid. We have changed this to state "the retrieval vertical and horizontal resolutions are inherently coupled together". The main idea we meant to convey is that crudely if we reduce the retrieval vertical resolution, there is more information available for the horizontal part.
* * *
**Page 7, lines 12-13** *"The effect of a one dimensional retrieval on horizontal regularization...."*

*- I guess you mean "horizontal resolution".*

**Reply**: Thank you, this has been changed.
* * *
**Table 1** *Please provide the reference tangent height for each interval.*

**Reply**: We have added the normalization altitude for each triplet to the table.
* * *
**Sect. 2.7.1** *What is the minimum retrieval altitude for ozone?*

**Reply**: This information is now available much earlier in the revised manuscript.
* * *
**Page 8, line 7** *Here and further below in the text you are talking about the "atmospheric upwelling". I suppose you mean the upwelling radiation. However, this notation is commonly used in the scientific community to describe the dynamic processes and means the upward moving air masses rather than radiance. Please use another notation throughout the text to avoid a confusion.*

**Reply**: We have changed all occurrences of "atmospheric upwelling" to "upwelling radiation".
* * *
**Page 8, lines 20-21** *I guess Eq. (4) is valid for both triplets and doublets. "... for triplet k " in line 21 should be "... for triplet l ".*

**Reply**: Corrected.
* * *
**Page 9** *"...any errors in the absolute calibration ..." - this is not completely true for an imaging instrument because the information for different tangent heights comes from different areas of the CCD and can have different calibration errors.*

**Reply**: This is true but we do not think our statement "helps to minimize any errors in the absolute calibration" contradicts this. The full line in the revised manuscript now reads "The high altitude normalization helps to minimize errors in the absolute calibration of the instrument and reduces the sensitivity to upwelling radiation."
* * *
**Page 10, Eq, (6)** *It is not clear how the second term is employed in the retrieval as the modeled Rayleigh background needs to be subtracted in the same way from both measured*

and modeled radiances and thus is canceled out when calculating $\mathbf{y} - F(\mathbf{x})$ in accordance with Eq. (2).

**Reply**: Thank you, this was confusing in the text. The Rayleigh subtraction is not used in the actual retrieval, although as you pointing out it would have no effect, it is only used to determine the high altitude normalization location based on the procedure of Bourassa et. al. 2012. We have modified the text to make this clear.
* * *
**Sect. 2.7.2** *No information is provided about how the aerosol extinction coefficient is calculated for other wavelengths.*

**Reply**: This is done using the same Mie code and assumed particle size distribution as for the phase function, the revised manuscript notes this and adds a reference to the source of the index of refraction data.
* * *
**Page 11, line 1** *"... albedo is handled in a two-dimensional sense ..."* - what is the second dimension for the albedo?

**Reply**: We have reworded the first part of this sentence to "While albedo in the forward model is allowed to vary in the horizontal direction".
* * *
**Sect. 2.7.3** *40 km tangent height to retrieve the surface albedo is quite high. Have you checked a possible influence of the stray light at this tangent height?*

**Reply**: We have done some internal tests here and have not noticed any significant effect on the retrieved ozone by changing the albedo retrieval height. Many past studies such as Loughman et al. 2005 have found that the absolute value of the retrieved albedo does not have a large effect on the ozone retrieval. Jaross et al. 2014 estimates the stray light contribution to be only 5% at 65 km for 750 nm, so we do not expect a large problem with using 40 km.
* * *
**Sect. 2.7.3** *The influence of the albedo spectral dependence must be discussed. For example, for a green vegetation the albedo obtained at 745 nm can be very different from that at 602 nm (red edge).*

**Reply**: We have added a reference to Loughman et. al. 2005 which discusses possible errors associated with neglecting the spectral albedo dependence. However it is important to remember that the albedo is not really surface reflection and is merely an approximation for the unknown diffuse upwelling radiation, thus you would not expect as harsh of a spectral

dependence that you would see with vegetation.
* * *
**Sect. 3** *The section is unnecessary as all discussed corrections are already implemented in the Level 1 v2.5 dataset of NASA.*

**Reply**: See our reply to the general comment above.
* * *
**Sect. 4** *If Levenberg-Marquardt term is used in the retrieval it must be also included in the error analysis.*

**Reply**: The Levenberg-Marquardt term does not appear in the cost function, and the error analysis is a linearization applied to the cost function so we do not see why this term should appear.
* * *
**Sect. 4** *Is the signal to noise of 100 is used only in the error analysis or in the standard retrieval as well? Why was not the signal to noise data provided in Level 1 data set used? The latter would provide a realistic instead of maximum error estimation.*

**Reply**: The SNR of 100 is used in both. The OMPS L1G documentation states that the SNR provided is an "Estimate of detector noise and not an estimate of random measurement uncertainty.", and we were told by the NASA OMPS-LP team that a value of 100 is more realistic.
* * *
**Page 13, line 9** *Only in the error analysis section the reader learn that the logarithm of the number density is the retrieval parameter rather than the number density itself. This must have been mentioned already in Sect. 2.2.*

**Reply**: The revised manuscript should correct this.
* * *
**Page 13, line 14** *what does "but near where the tropopause lowers at midlatitudes" refer to?*

**Reply**: Reworded to "near where the lower bound of the retrieval changes (due to the lowering tropopause) at mid-latitudes."
* * *
**Fig. 7** *Suboptimal color scale. How is the sign of the distance from the retrieval location defined?*

**Reply**: Color scale has been changed. The sign is negative towards the start the start of

the orbit in the southern hemisphere, we have added this to the figure caption.
* * *
**Page 14, line 3** *"Since the regularization term..." - once again, do not exclude the Levenberg-Marquardt term from the discussion.*

**Reply**: The Levenberg-Marquardt term does not have any effect here.
* * *
**Fig. 7** *The definition of the vertically/horizontally integrated averaging kernels is not quite clear. You have a set of averaging kernels for each vertical/horizontal grid point and each of them spans in both vertical and horizontal directions. Is the integration done over these directions? Is yes you seem to show one averaging kernel at each altitude in each panel in Fig. 7? If it was true I would expect the plural in the beginning of line 8 as you show multiple averaging kernels for different altitudes in each panel of Fig. 7. If my understanding of the definition is correct, I would like you to explain why there is a clear maximum at 40 km in tropics and 45 km at mid-latitudes and how it can be interpreted in terms of the retrieval sensitivity.*

**Reply**: As for the plural vs not plural, Technically there is only one averaging kernel for the entire orbit and what is being shown are multiple rows. We have modified the text to make this clear.

The difference in peak altitude is a little curious, we have two possible explanations. The first reason is that lines of sight below the tropopause are not used in the retrieval. At mid-latitudes we have lines of sight going from 10–18 km, which for the strongest absorbing UV triplets have peak sensitivity in the 40–50 km region owing to the optically thick line of sight path. Since these lines of sight are missing in the tropics the sensitivity peak is lower. Another way of thinking about this is that generally the information content is poorest at the retrieval boundaries, and increases away from them. Since the lower boundary is higher in altitude in the tropics it makes sense that the information maximum shifts downward.

The second cause is the difference in solar zenith angle between the two points. For the OMPS-LP geometry, the tropics have low solar zenith angles with minimal solar attenuation compared to the limb path. Higher latitudes have higher solar zenith angles where solar attenuation becomes more important. It is expected that sensitivity overall shifts upwards in areas with significant solar attenuation as the attenuation happens above the tangent point.
* * *
**Page 14, line 9** *"Only minor differences ..." - to my opinion the majority of differ- ences occur around 40 km and they are not minor.*

**Reply**: We see the referees point, however at 40 km the difference in FWHM is less than

25 km. We have reworded the text to state "Only minor differences in the FWHM ..."
* * *
**Page 14, lines 11-12** *"it was found that ..." - it is hard to believe as it is widely known that the averaging kernels for "relative" retrievals (i.e. retrieval of relative deviations from a priori or logarithms) depend on the atmospheric state. Please provide the averaging kernel plot for different season to justify you statement.*

**Reply**: We have added a second orbit (from a different season) averaging kernel to the figure. We have also reworded the offending sentence to "it was found that deviations from orbit to orbit are small enough that the above resolution estimates are representative for the entire dataset."
* * *
**Fig. 7** *why does the tropics plot have a white area below 18 km, how is the lower boundary of the retrieval defined?*

**Reply**: The data is masked below the lowest retrieval point which is the first altitude above the tropopause, we hope this is clear in the revised manuscript.
* * *
**Page 14, last paragraph** *It is absolutely inappropriate to neglect the instrument field of view without any investigations as it might lead to a significant change in both the retrieval results and error analysis.*

**Reply**: See the reply to the general comment.
* * *
**Sect. 5.1** *The results must be provided over the whole orbit as it is essential to estimate how the retrievals compare outside the vortex edge region. Another simulation for a different season with a vortex edge in the northern hemisphere needs to be provided to assess the influence of the viewing geometry.*

**Reply**: We have modified the first figure to show the entire orbit. We have also added a second simulation for a different season with a strong ozone gradient in the northern hemisphere.
* * *
**Page 15, line 25** *"For limb scatter measurements ..." - please illustrate this by plotting the averaging kernel for about 65° S and 15.5 km in both horizontal and vertical directions using a proper color scale.*

**Reply**: While the averaging kernel does also somewhat show this effect (in fact, it only does because some regularization is present) we do not think the requested figure is appropriate

to justify the statement "For limb scatter measurements ozone sensitivity is larger on the instrument side of the line of sight". The averaging kernel is specific to our 2D retrieval, and we are talking about an effect that is retrieval independent. The proper figure is one of the two-dimensional Jacobian for a single line of sight, of which there are many in paper referenced in the text (Zawada et. al. 2017).
* * *
**Sect. 5.2** *this section is not really informative and can be skipped. Details on the execution time suit better in the algorithm description section.*

**Reply**: We have removed this section and moved the information into the algorithm description section.
* * *
**Sect. 5.3** *Not only the anomalies but also the monthly mean values themselves need to be compared. This needs to be done for different latitude bands (tropics, mid-latitudes, high latitudes).*

**Reply**: See the reply to the general comment above.
* * *
**Fig. 10** *Why the altitudes above 59 km are not shown? If I understand it correctly, the retrieval runs up to 59 km.*

**Reply**: We assume the referee means why are altitudes above 50 km not shown. 50 km tends to be a common cutoff for ozone anomaly figures (and trend figures) due to the strong diurnal effect. In the coincident comparisons we can go above 50 km as the time difference between the two measurements is quite small. There are also issues where filtering MLS data according to the recommended procedure frequently cuts the data off ~55 km or occasionally lower, so below 50 km the sampling is roughly consistent for both instruments.
* * *
**Page 19, lines 1-3** *"... with the horizontal along-track resolution being poorer.."* - *please provide the values of the resolution and sampling for both instruments.*

**Reply**: We have added this information to the text.
* * *
**Page 19, lines 5-6** *".. has been degraded to the MLS pressure grid with a least square fit..."* - *please clarify what exactly was fitted and how you can degrade the vertical resolution using a least square fit. Here, a convolution with averaging kernels would be more suitable.*

**Reply**: This is simply the recommended procedure in the MLS data quality document, we have added a reference to the data quality document to indicate this. The OMPS-LP

measurements are converted to pressure at native resolution in pressure, then rather than interpolating these values to the MLS pressure grid a least squares fit is done assuming linear VMR variations. We have opted not to apply the MLS averaging kernels (in addition to the least squares fit) since our vertical resolution (estimated 1–2 km) is not significantly better than the MLS vertical resolution ($\sim$3 km). Futhermore, the MLS averaging kernels are fairly strongly peaked (peak values of 0.6 in the UTLS) so it would not be expected to make any significant differences. Jiang et al. 2007 did compare both of these methods (least squares fit vs averaging kernel) and found negligible differences even when comparing high resolution sonde measurements to a version of the MLS data with poorer vertical resolution than what we are using here.

Jiang, Y. B., et al. (2007), Validation of Aura Microwave Limb Sounder Ozone by ozonesonde and lidar measurements, J. Geophys. Res., 112, D24S34, doi:10.1029/2007JD008776.
* * *
**Fig. 12** *Provide the lower and upper altitude of the plotted range. Provide the same plot from 1D retrieval. Explain the lower limit of the retrieval.*

**Reply**: Tick labels have been added for the maximum and minimum of the plotted altitudes. The lower limit is now explained earlier in the revised manuscript. We do not see any value in adding results from the 1D retrieval here. Our only claim made about the 1D retrieval is that it has problems in the presence of large horizontal gradients, of which there are none in this orbit.
* * *
*It would be also interesting to show some examples from NASA Level 2 data, especially in Fig. 14.*

**Reply**: We agree this would be an interesting study, and in fact we believe there is work being done by other groups on comparing OMPS-LP retrievals by different processors, but we feel it is beyond the scope of this manuscript to include these comparisons.

**Technical corrections**

**Page 2, Eq. (1)** *matrices have to be shown in bold face to match the corresponding notations in the text.*

**Reply**: Corrected, thank you.
* * *
**Page 15, line 5** *duplicated word "those".*

**Reply**: Corrected, thank you.

---

## Referee Report (RR1)

**Referee report to "Tomographic retrievals of ozone with the OMPS Limb Profiler: algorithm description and preliminary results" by D. Zawada et al.**

Since the original submission the manuscript has been substantially improved especially in terms of presentation quality. A lot of details have been fixed and the manuscript reads much better now. However, the authors have refused to address the most crucial scientific deficits, as insufficient verification of the retrieval approach and method to calculate the aposteriori covariance and averaging kernels. To my opinion the paper cannot be published before the issues are addressed properly. Please find additional information in detailed comments below.

**Major issues:**

**Author's reply**: *The provided comparisons are intended to demonstrate the validity of the technique and not be a full validation of the dataset, which we feel is beyond the scope of this manuscript. As stated in the manuscript the validation work presented is preliminary, and a full validation is planned for a future paper.*

I do not agree that provided comparisons are sufficient to demonstrate the validity of the technique. The full comparison is provided only for one orbit and give no impression about a possible seasonal issues. Furthermore the amount of data is just too low. I also ask myself why Fig. 11 shows a comparison of anomalies instead of absolute values, which is quite unusual for a verification study. As there is almost no additional effort to show the same for the absolute values, I am really puzzled why the authors refuse to do that. I hope the reason is not to hide unexpected biases or mismatches in a seasonal behavior. I do not also think it is asked too much, to show similar plots for other latitudes. The differences would be sufficient.

**Author's reply**: *We do not believe this is correct in the standard use of the term "regularization". The Levenberg-Marquardt term does not appear in the cost function as would a traditional regularization term, and in theory, the retrieval should converge to the same solution (neglecting issues of multiple local minima) with or without the Levenberg-Marquardt term. It is true that the Levenberg-Marquardt term can have a regularization effect if the retrieval is stopped before proper convergence, but that is not the case here.*

As pointed out by Ceccherini and Ridolfi (2010) "In ill-conditioned retrievals the LM method acts as an external constraint and the solution depends on the path followed by the minimization procedure in the parameter space. This latter conclusion applies also to retrievals in which the iterations are stopped when a physically meaningful convergence criterion is fulfilled, i.e. before achievement of the numerical convergence at machine precision.". Even if we do not discuss the ill-conditioning, which is most always the case for limb profile retrieval, the latter condition is definitely the case for your algorithm (as described in the manuscript). Thus, the correct (as you would have them from the

numeric simulations) averaging kernel and the covariance matrices are not the same as when assuming the last iteration as a GN iteration (i.e. neglecting the Levenberg-Marquardt parameter). Due to this reason, the validity of the results presented in the "Error analysis and resolution" section is questionable.

Ceccherini, S. and Ridolfi, M.: Technical Note: Variance-covariance matrix and averaging kernels for the Levenberg-Marquardt solution of the retrieval of atmospheric vertical profiles, Atmos. Chem. Phys., 10, 3131-3139, https://doi.org/10.5194/acp-10-3131-2010, 2010.

**Author's reply**:*The actual field of view (both magnitude and shape) varies as a function of altitude and wavelength depending on where each pixel is, and this information is not publicly available. Furthermore, neglecting the instrumental field of view is a common assumption in many limb retrievals (including the operational NASA OMPS-LP retrieval and the OSIRIS retrieval, which this work builds upon) and we do not feel it is within the scope of this manuscript to perform a full study on this effect.*

The information on the effective field of view is provided in Level 1 ATBD and can be used to make at least a rough estimation whether the effect is significant. Am not aware of the fact that " neglecting the instrumental field of view is a common assumption in many limb retrievals". This has to be confirmed by citations. The operational NASA algorithm is not yet published as far as I know, so it is too early to discuss it. The OSIRIS algorithm is run by the same research group and cannot be cited as a "common standard".

**Minor issues:**

- Page 4, line 6: It not clear how 5° is calculated. May be it is better just to skip the angle and use the distance as comes thereafter.

- Page 5, line 31: "cone" term - Once again, cone is per definition a three dimensional shape. As long as you work with two spatial coordinates (altitude and angle determining the orbital position) the usage of the term "cone" is inappropriate. It is rather a sector.

- Page 7, Eq. (3): The domain of the sums is not defined. I guess the first sum runs other different reference wavelengths, which is over 1 at least for the Chappuis band, but I see no point for the second sum. What are you going to sum up here?

- Page 7: "helps to minimize errors in the absolute calibration of the instrument" - still, it is only true if the errors are the same for both tangent heights. Otherwise the errors might be even amplified. Unfortunately, we do not know which is the case for OMPS.

- Sect. 2.8.1: The description of the aerosol retrieval is still confusing. I suggest to remove the paragraph after Eq. (5) writing instead that the reference tangent height is selected in accordance with (Bourassa et al. 2012).

- Fig. 14 (former Fig. 12): In their reply to my comment authors write "We do not see any value in adding results from the 1D retrieval here. Our only claim made about the 1D retrieval is that it has problems in the presence of large horizontal gradients, of which there are none in this orbit." - I think the authors miss here an important point. While the manuscript is focused on demonstration advantages of 2D retrieval (and this is fine) the authors do not care about demonstrating the fact that 2D retrieval does not decrease the retrieval quality outside the polar vortex regions, e.g. by smoothing out some horizontal features. From this point of view a value of providing similar plot for 1D retrieval would be to demonstrate that the overall performance does not get worse, which is also a very important finding related to the algorithm quality assessment.

**Technical comments:**

- Page 2, line 21: "at select" $\longrightarrow$ "at selected"

- Page 2, Eq. (1): "F" should be either bold in the equation or regular in the line below

- Page 2, last line: extra symbol between "0.1" and "the mean"

---

## Referee Report (RR2)

**Referee report to "Tomographic retrievals of ozone with the OMPS Limb Profiler: algorithm description and preliminary results" by D. Zawada et al.**

I appreciate the extended validation and some text adjustments made by authors in accordance with my comments. Although the answers to the remaining comments are still not fully convincing I agree to close the discussion with respect to the majority of the issues as they are rather minor and do not justify the decision to do not accept the paper. However, two of the remaining issues (as listed below) I still consider major and recommend to sort them out before the manuscript can be accepted for publication.

**Major issues:**

**Author's reply**: ... *We choose to show anomalies for two reasons. The first is that the primary use of the dataset so far is in trend analyses, where the anomalies are used. Second, tropical anomalies are shown to demonstrate that features such as the anomalous QBO disruption are present in the dataset. Anomalies for other latitude bands and the seasonal cycle of the dataset have already been studied in detail by Sofieva et. al. (2017), and we do not see the value in repeating similar analyses here....*

May be the authors misunderstood my comment, I do not require to replace the anomaly plot by the plot with absolute values. I just would like you to add another plot showing the differences in absolute values. As you already have all the data it requires, to my opinion, just a minor effort. I agree that the anomalies are very interesting to see, but the agreement in the absolute values is also of a great interest and I cannot understand the reasons why you refuse to show it. The reference to Sofieva et. al. (2017) is irrelevant in this respect, as this paper does not consider MLS data, which your comparisons are focused at.

**Author's reply**: *We agree that the atmospheric inverse problem is typically ill-conditioned, but this is only without regularization present. The purpose of including regularization is to improve the conditioning of the problem so the first statement does not apply to our retrieval. It is true that including the LM term changes the path to the solution, but we do not agree that this changes the solution. In a comment on the discussion version of the Ceccherini et al. paper (https://www.atmos-chem-phys-discuss.net/9/C9660/2010/acpd-9-C9660-2010.pdf ) Dr. von Clarmann argues that "converged Levenberg-Marquardt retrievals are characterized by the same covariance matrices and averaging kernels as Gauss-Newton or optimal estimation retrievals." In the reply to this comment by Ceccherini et al. (https://www.atmos-chem- phys-discuss.net/9/C10551/2010/acpd-9-C10551-2010.pdf ) they state "The differences that exist between our results and the expectations of Dr. von Clarmann can be explained by the fact that the test retrievals presented in the discussion paper do not use external constraints (R=0)" and furthermore that "In the case of well conditioned problems the formulas of the discussion paper produce the same results as those of Dr. von Clarmann." Therefore it is not correct to include the Levenberg-Marquardt term*

*in the error analysis and characterization.*

My original comment did not refer to the solution. The issue was that the method to calculate the averaging kernels and solution variances is to my opinion not quite correct. In this respect, I see no contradiction between the findings published in the final paper by Ceccherini and Ridolfi (2010) and comments by Dr. von Clarmann to the discussion paper. Considering the issues listed below, both documents also perfectly conform to my comment.

- Dr. von Clarmann in his comments starts from Eq.(2) to arrive to the conclusion that "converged Levenberg-Marquardt retrievals are characterized by the same covariance matrices and averaging kernels as Gauss-Newton or optimal estimation retrievals". However, the discussion flow and the conclusion are only valid if the matrix $\vec{R}$ ($R^T R$ in the notations of the manuscript) in Eq.(2) in invertible. This is however not the case for the regularization matrix given by Eq. (7) of the manuscript.

- As pointed out by Dr. von Clarmann in his comments "With a large Levenberg-Marquardt term it is easy to obtain a small relative variation of the chi-square although the retrieval is still far from convergence, and retrievals where the iteration has been interrupted without making sure that $\vec{x}_{i+1} - \vec{x}_i \to 0$ even without a damping term should be discarded and by no means be accepted as a solution". In Sect. 2.6 of the manuscript the authors state that they do a fixed number of iterations and analyze only the chi-square. Thus, it is highly probable that the retrieval is non-converged and the situation highlighted by Ceccherini and Ridolfi (2010) occurs: "... the LM method acts as an external constraint and the solution depends on the path followed by the minimization procedure in the parameter space. This latter conclusion applies also to retrievals in which the iterations are stopped when a physically meaningful convergence criterion is fulfilled, i.e. before achievement of the numerical convergence at machine precision"

Thus, to my opinion, the study by Ceccherini and Ridolfi (2010) is highly relevant for the retrieval used in the manuscript and the method to calculate the averaging kernels and variances described by Ceccherini and Ridolfi (2010) needs to be applied. Furthermore, authors should reconsider their iterative approach to ensure that the retrieval converges.

Ceccherini, S. and Ridolfi, M.: Technical Note: Variance-covariance matrix and averaging kernels for the Levenberg-Marquardt solution of the retrieval of atmospheric vertical profiles, Atmos. Chem. Phys., 10, 3131-3139, https://doi.org/10.5194/acp-10-3131-2010, 2010.

**Minor comments:**

Page 19, line 33: "upwelling" $\rightarrow$ "upwelling radiation"

Page 21, line 7: Please make a statement whether the selected 251 orbits are evenly distributed over the seasons or not.

Page 21, lines 13 - 14: "The generally good agreement ..." $\rightarrow$ it should be noted here that the observed standard deviation includes the natural variability while the predicted one does not. So, the good agreement between these two values is not necessary a good sign.

---

## Referee Report (RR3)

**Referee report to "Tomographic retrievals of ozone with the OMPS Limb Profiler: algorithm description and preliminary results" by D. Zawada et al.**

I appreciate the efforts made by the authors to address the remaining major issues in an appropriate way. With respect to the newly written/changed parts of the text the are still a couple of minor issues/technical corrections need to be addressed. My detailed comments are listed below. I recommend to accept the paper for publishing in AMT after the listed issues have been addressed.

**Minor comments:**

Page 20, lines 15-16: "..., which are expected as measurements from OMPS-LP consist of purely stratospheric air while measurements from MLS are a combination of stratospheric and tropospheric air." $\rightarrow$ it is unclear how the statement is justified and if it is correct. Do you mean a different sensitivity to clouds, differences in the vertical resolution or something else? The statement needs to be explained in more details or deleted.

Page 20, lines 16-18: "Above 45 km OMPS-LP is low relative to MLS, which is explained through the diurnal cycle of upper stratospheric ozone. OMPS-LP measures during the day, catching only the low part of the cycle, while MLS measures both day and night." $\rightarrow$ With this statements authors might give the reader an impression that they do not know, that day- and nighttime measurements should be considered separately when performing comparisons for photochemically active species. As the author team has an extensive experience in the retrieval and validation I am sure this is not a lack of understanding but is done just for a sake of simplicity, as this comparison is rather outside the main focus of the paper. Generally, I would agree with this strategy if the authors stated clearly that they recognize the comparison is "quick and dirty" at this point but they think it is unnecessary to do more efforts. This can be done, for example by replacing the last two sentences of the section by "The comparison results above 45 km are not representative, as both day- and nighttime measurements of MLS are used to calculate the monthly zonal mean values, which is inappropriate when the diurnal variation of ozone becomes significant."

**Technical corrections:**

Page 6, lines 23-24: "To verify that convergence has been reached, at every iteration both the current $\chi^2$ value and the expected $\chi^2$ value at the next step assuming the problem is linear." $\rightarrow$ the sentence seems to be incomplete.

Page 14, lines 15-16: "At the end of the retrieval are near identical to unity with peak values of 0.99 in the worst case." $\rightarrow$ noun is missing.

---

## Author Response (AR2)

**Author's response for amt-2017-236**

Daniel Zawada et al.

January 23, 2018

We would like to thank the editor for handling the manuscript and the reviewers for their helpful comments. What follows is a point by point response to each of the reviewers suggestions as well as a marked up version of the changes in the revised manuscript.

**Response to Referee 1 (Alexei Rozanov)**

Daniel J. Zawada, Landon A. Rieger, Adam E. Bourassa, and Douglas A. Degenstein

January 23, 2018

We would like to thank the referee for their helpful comments and suggestions. Included below is each of the referee's comments (italics) followed by our reply.
* * *
*I do not agree that provided comparisons are sufficient to demonstrate the validity of the technique. The full comparison is provided only for one orbit and give no impression about a possible seasonal issues. Furthermore the amount of data is just too low. I also ask myself why Fig. 11 shows a comparison of anomalies instead of absolute values, which is quite unusual for a verification study. As there is almost no additional effort to show the same for the absolute values, I am really puzzled why the authors refuse to do that. I hope the reason is not to hide unexpected biases or mismatches in a seasonal behavior. I do not also think it is asked too much, to show similar plots for other latitudes. The differences would be sufficient.*

**Reply**: As stated in the manuscript, the main, important result presented here is the 2D tomographic inversion. We are not attempting to provide a full validation of the OMPS ozone product; we are explaining the retrieval approach and showing that it works. However, we have extended the comparison with MLS to now include all 251 coincident orbits from 2012-2013 in a new version of Fig 14. This includes mean difference in various latitude bands. We choose to show anomalies for two reasons. The first is that the primary use of the dataset so far is in trend analyses, where the anomalies are used. Second, tropical anomalies are shown to demonstrate that features such as the anomalous QBO disruption are present in the dataset. Anomalies for other latitude bands and the seasonal cycle of the dataset have already been studied in detail by Sofieva et. al. (2017), and we do not see the value in repeating similar analyses here. A citation has been added to Sofieva et al. (2017) referencing the anomaly comparisons in other latitude bands.

Sofieva, V. F., Kyrola, E., Laine, M., Tamminen, J., Degenstein, D., Bourassa, A., Roth, C., Zawada, D., Weber, M., Rozanov, A., Rahpoe, N., Stiller, G., Laeng, A., von Clarmann, T., Walker, K. A., Sheese, P., Hubert, D., van Roozendael, M., Zehner, C., Damadeo, R., Zawodny, J., Kramarova, N., and Bhartia, P. K.: Merged SAGE II, Ozone_cci and OMPS ozone profile dataset and evaluation of ozone trends in the stratosphere, Atmos. Chem.

Phys., 17, 12533-12552, https://doi.org/10.5194/acp-17-12533-2017, 2017.
* * *
*As pointed out by Ceccherini and Ridolfi (2010) "In ill-conditioned retrievals the LM method acts as an external constraint and the solution depends on the path followed by the minimization procedure in the parameter space. This latter conclusion applies also to retrievals in which the iterations are stopped when a physically meaningful convergence criterion is fulfilled, i.e. before achievement of the numerical convergence at machine precision.". Even if we do not discuss the ill-conditioning, which is most always the case for limb profile retrieval, the latter condition is definitely the case for your algorithm (as described in the manuscript). Thus, the correct (as you would have them from the numeric simulations) averaging kernel and the covariance matrices are not the same as when assuming the last iteration as a GN iteration (i.e. neglecting the Levenberg-Marquardt parameter). Due to this reason, the validity of the results presented in the "Error analysis and resolution" section is questionable.*

*Ceccherini, S. and Ridolfi, M.: Technical Note: Variance-covariance matrix and averaging kernels for the Levenberg-Marquardt solution of the retrieval of atmospheric vertical profiles, Atmos. Chem. Phys., 10, 3131-3139, https://doi.org/10.5194/acp-10-3131-2010, 2010.*

**Reply**: We agree that the atmospheric inverse problem is typically ill-conditioned, but this is only without regularization present. The purpose of including regularization is to improve the conditioning of the problem so the first statement does not apply to our retrieval. It is true that including the LM term changes the path to the solution, but we do not agree that this changes the solution. In a comment on the discussion version of the Ceccherini et al. paper (https://www.atmos-chem-phys-discuss.net/9/C9660/2010/acpd-9-C9660-2010.pdf) Dr. von Clarmann argues that "converged Levenberg-Marquardt retrievals are characterized by the same covariance matrices and averaging kernels as Gauss-Newton or optimal estimation retrievals." In the reply to this comment by Ceccherini et al. (https://www.atmos-chem-phys-discuss.net/9/C10551/2010/acpd-9-C10551-2010.pdf) they state "The differences that exist between our results and the expectations of Dr. von Clarmann can be explained by the fact that the test retrievals presented in the discussion paper do not use external constraints (R=0)" and furthermore that "In the case of well conditioned problems the formulas of the discussion paper produce the same results as those of Dr. von Clarmann." Therefore it is not correct to include the Levenberg-Marquardt term in the error analysis and characterization.
* * *
*The information on the effective field of view is provided in Level 1 ATBD and can be used to make at least a rough estimation whether the effect is significant. Am not aware of the fact that " neglecting the instrumental field of view is a common assumption in many limb retrievals". This has to be confirmed by citations. The operational NASA algorithm is not yet published as far as I know, so it is too early to discuss it. The OSIRIS algorithm is run by the same research group and cannot be cited as a "common standard".*

**Reply**: While the OSIRIS algorithm may not be a "common standard" it is an established technique. As this work builds upon many aspects of the OSIRIS algorithm (use of SASKTRAN, definition of measurement vectors) we do not believe it is within the scope of the manuscript quantify already established assumptions. We stated in our previous reply that preliminary simulation tests have indicated that using 1.5 km as the field of view causes biases $<2\%$ primarily in the 20–25 km region. We have added a statement of this effect to the manuscript. As a sidenote we would like to point out that a previous version of the operational NASA algorithm is published by Rault and Loughman (2013), and while this publication does not explicitly mention neglecting the field of view the ATBD (https://ozoneaq.gsfc.nasa.gov/media/docs/EDR_ATBD_baseline_version1.pdf), of which the publication is based upon, does in section 3.6.
* * *
**Page 4, line 6** *It not clear how 5° is calculated. May be it is better just to skip the angle and use the distance as comes thereafter.*

**Reply**: The 5° can be calculated either directly from the OMPS-LP tangent point data or a simple calculation knowing the orbital period and the rotation of the Earth. We would prefer to leave both quantities in as it provides an easy way to visualize the effect.
* * *
**Page 5, line 31** *"cone" term - Once again, cone is per definition a three dimensional shape. As long as you work with two spatial coordinates (altitude and angle determining the orbital position) the usage of the term "cone" is inappropriate. It is rather a sector.*

**Reply**: At this point we are talking about SASKTRAN-HR solving the radiative transfer equation, not the retrieval. By solving the radiative transfer equation we are referring to calculating the source function, which is 5 dimensional with 3 spatial dimensions. The term cone corresponds to the 3 spatial dimensions in which the source function is calculated. To make this clear we have reworded "Since SASKTRAN-HR solves the radiative transfer equation ..." to "Since SASKTRAN-HR solves the source function of the radiative transfer equation ...".
* * *
**Page 7, Eq. (3)** *The domain of the sums is not defined. I guess the first sum runs other different reference wavelengths, which is over 1 at least for the Chappuis band, but I see no point for the second sum. What are you going to sum up here?*

**Reply**: As requested we have removed the second sum.
* * *
**Page 7** *"helps to minimize errors in the absolute calibration of the instrument" - still, it is only true if the errors are the same for both tangent heights. Otherwise the errors might be even amplified. Unfortunately, we do not know which is the case for OMPS.*

**Reply**: We agree that certain types of calibration errors will not be reduced by high altitude

normalization since OMPS-LP is an imaging system, however there are errors that will be. Jaross et al. (2014) state that there are systematic differences between the two different apertures of OMPS-LP, and because of this it was decided to calculate each vertical radiance profile at a single wavelength from a single aperture and not both. The systematic error that causes this difference is an example of a systematic error that could be reduced by high altitude normalization. We do not believe that our statement is false, and that it does not preclude situations described by the referee where errors may increase.
* * *
**Sect. 2.8.1** *The description of the aerosol retrieval is still confusing. I suggest to remove the paragraph after Eq. (5) writing instead that the reference tangent height is selected in accordance with (Bourassa et al. 2012).*

**Reply**: Changed as suggested.
* * *
**Fig. 14 (former Fig. 12)** *In their reply to my comment authors write "We do not see any value in adding results from the 1D retrieval here. Our only claim made about the 1D retrieval is that it has problems in the presence of large horizontal gradients, of which there are none in this orbit." - I think the authors miss here an important point. While the manuscript is focused on demonstration advantages of 2D retrieval (and this is fine) the authors do not care about demonstrating the fact that 2D retrieval does not decrease the retrieval quality outside the polar vortex regions, e.g. by smoothing out some horizontal features. From this point of view a value of providing similar plot for 1D retrieval would be to demonstrate that the overall performance does not get worse, which is also a very important finding related to the algorithm quality assessment.*

**Reply**: We believe the referee intends to refer to Fig. 13 rather than Fig. 14, as that is Fig. 12 in the prior submission.

We agree that smoothing of horizontal features by both the 2D and 1D retrievals is an interesting topic and something that could be looked into more detail in the future. However we believe that we have both quantified this effect for the 2D retrieval through the averaging kernel, and also demonstrated this in practice through simulated retrievals. We do not think that including the 1D retrieval in Fig. 13 would demonstrate that the "[2D retrieval] overall performance does not get worse" as we are comparing to MLS, which has a similar horizontal

smoothing constraint.
* * *
**Page 2, line 21** *"at select"* → *"at selected"*

**Reply**: Thank you, Changed
* * *
**Page 2, Eq. (1)** *"F" should be either bold in the equation or regular in the line below*

**Reply**: Thank you, we have changed it to bold in the equation.
* * *
**Page 2, last line** *extra symbol between "0.1" and "the mean"*

**Reply**: Thank you, this was a multiplication symbol that looked funny. We have changed this to "$\gamma_i = 0.1$ multiplied by the mean ..."

[revised manuscript text omitted]

---

## Author Response (AR3)

**Author's response for amt-2017-236**

Daniel Zawada et al.

March 19, 2018

We would like to thank the editor for handling the manuscript and the reviewer for their helpful comments. What follows is a point by point response to each of the reviewers suggestions as well as a marked up version of the changes in the revised manuscript.

**Response to Referee 1 (Alexei Rozanov)**

We would like to thank the referee for their helpful comments and suggestions. Included below is each of the referee's comments (italics) followed by our reply.
* * *
*Maybe the authors misunderstood my comment, I do not require to replace the anomaly plot by the plot with absolute values. I just would like you to add another plot showing the differences in absolute values. As you already have all the data it requires, to my opinion, just a minor effort. I agree that the anomalies are very interesting to see, but the agreement in the absolute values is also of a great interest and I cannot understand the reasons why you refuse to show it. The reference to Sofieva et. al. (2017) is irrelevant in this respect, as this paper does not consider MLS data, which your comparisons are focused at.*

**Reply**: As requested we have added a new figure to the revised manuscript which shows the absolute value comparison for the same case.
* * *
*My original comment did not refer to the solution. The issue was that the method to calculate the averaging kernels and solution variances is to my opinion not quite correct. In this respect, I see no contradiction between the findings published in the final paper by Ceccherini and Ridolfi (2010) and comments by Dr. von Clarmann to the discussion paper. Considering the issues listed below, both documents also perfectly conform to my comment.*

- *Dr. von Clarmann in his comments starts from Eq.(2) to arrive to the conclusion that "converged Levenberg-Marquardt retrievals are characterized by the same covariance matrices and averaging kernels as Gauss-Newton or optimal estimation retrievals". However, the discussion flow and the conclusion are only valid if the matrix $R$ ($R^T R$ in the notations of the manuscript) in Eq.(2) in invertible. This is however not the case for the regularization matrix given by Eq. (7) of the manuscript.*

- *As pointed out by Dr. von Clarmann in his comments "With a large Levenberg-Marquardt term it is easy to obtain a small relative variation of the chi-square although the retrieval is still far from convergence, and retrievals where the iteration has been interrupted without making sure that $x_{i+1} - x_i \to 0$ even without a damping term should be discarded and by no means be accepted as a solution". In Sect. 2.6 of the manuscript the authors state that they do a fixed number of iterations and analyze only the chi-square. Thus, it is highly probable that the retrieval is non-converged and the situation highlighted by Ceccherini and Ridolfi (2010) occurs: "... the LM method*

*acts as an external constraint and the solution depends on the path followed by the minimization procedure in the parameter space. This latter conclu- sion applies also to retrievals in which the iterations are stopped when a physically meaningful convergence criterion is fulfilled, i.e. before achievement of the numerical convergence at machine precision"*

*Thus, to my opinion, the study by Ceccherini and Ridolfi (2010) is highly relevant for the retrieval used in the manuscript and the method to calculate the averaging kernels and variances described by Ceccherini and Ridolfi (2010) needs to be applied. Furthermore, authors should reconsider their iterative approach to ensure that the retrieval converges.*

*Ceccherini, S. and Ridolfi, M.: Technical Note: Variance-covariance matrix and averaging kernels for the Levenberg-Marquardt solution of the retrieval of atmospheric vertical profiles, Atmos. Chem. Phys., 10, 3131-3139, https://doi.org/10.5194/acp-10-3131-2010, 2010.*

**Reply**: In regards to the first point, it can be seen that for a simple linear system Eq. (2) can be solved directly for the state vector without inverting $R$. We acknowledge there may be some ambiguity when the system is not linear, however the second part of Dr. von Clarmann's comment which shows that the method of Ceccherini et al. converges to standard forms of the gain matrix does not require $R$ to be inverted.

For the second point, we apologize if there was confusion in the text, however we are not only comparing $\chi^2$ values from each iteration. We agree that comparing $\chi^2$ values that include an LM term can cause the retrieval to appear converged when it is not. What we had stated in the manuscript was "Various diagnostic information is also calculated, including the normalized $\chi^2$ value and the expected $\chi^2$ value at the next step assuming the problem is linear. At the end of the fixed number of iterations it was found that these two values always match within 1%, indicating that the solution has likely converged." By expected $\chi^2$ value at the minimum assuming the problem is linear we are referring to an estimate calculated with the LM term explicitly set to 0 to guard against the type of situation the reviewer described. A similar technique is done in Livesey et al. (2006) for the operational MLS retrievals. We have expanded the description of the post retrieval convergence sanity checks in the revised manuscript to hopefully clarify this issue.

Lastly, to check whether or not the methodology of Ceccherini and Ridolfi (2010) is required for the retrieval presented we have added a short study to the manuscript. The vertical averaging kernel is calculated using the methodology of Ceccherini and Ridolfi (2010) for a subset of an OMPS-LP orbit and was found to be essentially unity (peak values of 0.99 in the worst case). Therefore the retrieval is sufficiently converged that applying the methodology of Ceccherini and Ridolfi (2010) is not necessary.

Livesey, N. J., Van Snyder, W., Read, W. G., & Wagner, P. A. (2006). Retrieval algorithms for the EOS Microwave limb sounder (MLS). IEEE transactions on geoscience and

remote sensing, 44(5), 1144-1155.
* * *
**Page 19, line 33** *"upwelling"* → *"upwelling radiation"*

**Reply**: Changed, thank you. We also changed a few other occurences of similar statements.
* * *
**Page 21, line 7** *Please make a statement whether the selected 251 orbits are evenly distributed over the seasons or not.*

**Reply**: We have added a statement that the orbits are uniformly distributed over the time period.
* * *
**Page 21, lines 13–14** *"The generally good agreement ..."* → *it should be noted here that the observed standard deviation includes the natural variability while the predicted one does not. So, the good agreement between these two values is not necessary a good sign.*

**Reply**: We agree that the observed standard deviation does include natural variability, however the amount of natural variability here should be negligible with the tight coincidence criteria that is used (less than 20 minutes, all measurements within 1° longitude). We have added the statement "
[revised manuscript text omitted]

---

## Author Response (AR4)

**Author's response for amt-2017-236**

Daniel Zawada et al.

April 9, 2018

We would like to thank the editor for handling the manuscript and the reviewer for their helpful comments. What follows is a point by point response to each of the reviewers suggestions as well as a marked up version of the changes in the revised manuscript.

**Response to Referee 1 (Alexei Rozanov)**

We would like to thank the referee for their helpful comments and suggestions. Included below is each of the referee's comments (italics) followed by our reply.
* * *
**Page 20, Lines 15–16** *". . . , which are expected as measurements from OMPS-LP consist of purely stratospheric air while measurements from MLS are a combination of stratospheric and tropospheric air."* → *it is unclear how the statement is justified and if it is correct. Do you mean a different sensitivity to clouds, differences in the vertical resolution or something else? The statement needs to be explained in more details or deleted.*

**Reply**: We agree this could be explained a bit better. Here we are referring to a sampling bias that arises due to the OMPS-LP retrieval lowerbound being set to the tropopause. This means that any at low altitudes the montly zonal mean value from OMPS-LP consists entirely of measurements above the tropopause, while the same value calculated for MLS contains measurements that were both above and below the tropopause. Since we expect the amount of ozone to be generally higher above the tropopause this is a possible explanation for the high bias that we see. We have reworded this statement to "Large high biases can be seen at the lowest altitudes (16–18 km), which could be explained by the retrieval lowerbound being set to the tropopause causing a sampling bias. Values used to calculate the monthly zonal mean for MLS could consist of both tropospheric and stratospheric air, while monthly zonal means from OMPS-LP are purely stratospheric leading to higher observed values."
* * *
*"Above 45 km OMPS-LP is low relative to MLS, which is explained through the diurnal cycle of upper stratospheric ozone. OMPS-LP measures during the day, catching only the low part of the cycle, while MLS measures both day and night."* → *With this statements authors might give the reader an impression that they do not know, that day- and nighttime measurements should be considered separately when performing comparisons for photochemically active species. As the author team has an extensive experience in the retrieval and validation I am sure this is not a lack of understanding but is done just for a sake of simplicity, as this comparison is rather outside the main focus of the paper. Generally, I would agree with this strategy if the authors stated clearly that they recognize the comparison is "quick and dirty" at this point but they think it is unnecessary to do more efforts. This can be done, for example by replacing the last two sentences of the section by "The comparison results above 45 km are not representative, as both day- and nighttime measurements of*

MLS are used to calculate the monthly zonal mean values, which is inappropriate when the diurnal variation of ozone becomes significant."

**Reply**: Thank you, you are correct that this was done for the case of simplicity. We have modified the last two sentences to read "Above 45 km OMPS-LP is low relative to MLS, however these results are not representative as both day and nighttime measurements are used to calculate the MLS monthly zonal means, which causes differences when the diurnal variation of ozone is significant."
* * *
**Page 6, lines 23–24** *"To verify that convergence has been reached, at every iteration both the current $\chi^2$ value and the expected $\chi^2$ value at the next step assuming the problem is linear."* $\rightarrow$ *the sentence seems to be incomplete.*

**Reply**: Thank you, "are calculated" was missing from the end of the sentence.
* * *
**Page 14, lines 15–16** *"At the end of the retrieval are near identical to unity with peak values of 0.99 in the worst case."* $\rightarrow$ *noun is missing.*

[revised manuscript text omitted]